# ETNK1 mutations induce a mutator phenotype that can be reverted with phosphoethanolamine

Diletta Fontana [1], Mario Mauri[1], Rossella Renso[1], Mattia Docci[1], Ilaria Crespiatico[1], Lisa M. Røst [2], Mi Jang[2], Antonio Niro[1], Deborah D'Aliberti[1], Luca Massimino [1], Mayla Bertagna[1], Giovanni Zambrotta[1], Mario Bossi[1], Stefania Citterio[3], Barbara Crescenzi[4], Francesca Fanelli[5,6], Valeria Cassina [1], Roberta Corti[1], Domenico Salerno[1], Luca Nardo[1], Clizia Chinello[1], Francesco Mantegazza [1], Cristina Mecucci [4], Fulvio Magni[1], Guido Cavaletti [1], Per Bruheim [2], Delphine Rea[7], Steen Larsen [8,9], Carlo Gambacorti-Passerini [1,10,12] & Rocco Piazza [1,10,11,12✉]

Recurrent somatic mutations in *ETNK1* (Ethanolamine-Kinase-1) were identified in several myeloid malignancies and are responsible for a reduced enzymatic activity. Here, we demonstrate in primary leukemic cells and in cell lines that mutated ETNK1 causes a significant increase in mitochondrial activity, ROS production, and Histone H2AX phosphorylation, ultimately driving the increased accumulation of new mutations. We also show that phosphoethanolamine, the metabolic product of ETNK1, negatively controls mitochondrial activity through a direct competition with succinate at mitochondrial complex II. Hence, reduced intracellular phosphoethanolamine causes mitochondria hyperactivation, ROS production, and DNA damage. Treatment with phosphoethanolamine is able to counteract complex II hyperactivation and to restore a normal phenotype.

[1] Department of Medicine and Surgery, University of Milano - Bicocca, Monza, Italy. [2] Department of Biotechnology and Food Science, Norwegian University of Science and Technology, Trondheim, Norway. [3] Department of Biotechnology and Biosciences, University of Milano - Bicocca, Milano, Italy. [4] Centro Ricerche Emato-Oncologiche, University of Perugia, Perugia, Italy. [5] Department of Life Sciences, University of Modena and Reggio Emilia, Modena, Italy. [6] Center for Neuroscience and Neurotechnology, University of Modena and Reggio Emilia, Modena, Italy. [7] Service d'Hématologie adulte, Hôpital Saint-Louis, Paris, France. [8] X-lab, Center for Healthy Aging, Department of Biomedical Sciences, University of Copenhagen, Copenhagen, Denmark. [9] Clinical Research Centre, Medical University of Bialystok, Bialystok, Poland. [10] Hematology and Clinical Research Unit, San Gerardo Hospital, Monza, Italy. [11] Bicocca Bioinformatics, Biostatistics and Bioimaging Centre (B4), University of Milano - Bicocca, Milan, Italy. [12] These authors contributed equally: Carlo Gambacorti-Passerini, Rocco Piazza. ✉email: rocco.piazza@unimib.it

ETNK1 kinase is responsible for the phosphorylation of ethanolamine (Et) to phosphoethanolamine (P-Et). P-Et plays a critical role in the Kennedy pathway, representing the main metabolic route by which mammalian cells synthesize the two most abundant phospholipids of the cell membrane: phosphatidylethanolamine (PE) and phosphatidylcholine (PC). Specifically, P-Et synthesis represents the first metabolic step required for PE anabolism[1].

PE plays a major role in defining cell membrane architecture[2]. It is believed to cause lateral pressure and to introduce curvature stress, therefore influencing folding and activity of several membrane proteins[3]. Its presence is critical at the end of the cytokinesis, where cells lacking PE are unable to complete the cell division process[4]. Finally, PE is required for an optimal mitochondrial respiratory activity and ubiquinone function[5].

By using Next Generation Sequencing techniques, we and others identified recurrent missense somatic mutations occurring on ETNK1 in about 13% of patients affected by atypical chronic myeloid leukemia (aCML)[6], in 3–14% of chronic myelomonocytic leukemia (CMML)[6,7], and in 20% of systemic mastocytosis (SM) patients with eosinophilia[7]. Following these findings, ETNK1 mutations were included in the World Health Organization (WHO) 2016 classification as a support criterion for the diagnosis of aCML[8]. ETNK1 mutations, encoding for H243Y, N244S/T/K, and G245V/A amino acid substitutions, cluster in a very narrow region of the ETNK1 catalytic domain and cause an impairment of ETNK1 enzymatic activity leading to a significant decrease in the intracellular concentration of P-Et[6]. Recently, somatic ETNK1 mutations occurring in the same mutational hotspot were also described in diffuse large B-cell lymphomas (DLBCL)[9], supporting the notion that these mutations are not restricted to myeloid disorders.

Here, we investigate the specific role of these mutations by using cellular CRISPR/Cas9 and ETNK1 overexpression models as well as patient samples. We show that ETNK1 mutations are responsible for mitochondria hyperactivation owing to a direct competition between P-Et and succinate for mitochondrial complex II succinate dehydrogenase (SDH). In turn, mitochondria hyperactivation leads to increased ROS production and to the induction of a mutator phenotype. We also show that treatment with P-Et is able to fully counteract this process.

## Results

**ETNK1 mutations increase mitochondria activity.** To study the biological effect of ETNK1 mutations we generated CRISPR/Cas9 models of mutated (ETNK1-N244S) and knock-out (ETNK1-KO) ETNK1 on the HEK293-Flp-In cell line (Supplementary Data 1). CRISPR/Cas9 clones were validated using targeted sequencing (Supplementary Fig. 1), FISH (see "Methods" section for further details), and quantitative real-time PCR (Supplementary Fig. 2).

As the presence of a physiological PE concentration in mitochondria membranes is reported to be critical for the oxidative phosphorylation pathway[10,11], we investigated mitochondria respiratory chain activity. Analyses done on target cells by using MitoTracker Red and Green to assess mitochondria potential and mass showed an absolute increase of mitochondrial mass (Fig. 1a; 1.38 and 1.33 fold increase in ETNK1-N244S and ETNK1-KO compared to ETNK1-WT; $p = 0.0099$ and $p = 0.0254$, respectively) and activity in both mutated and knock-out ETNK1 lines (Fig. 1b; 1.87 and 2.48 fold increase in ETNK1-N244S and ETNK1-KO compared to ETNK1-WT; $p = 0.0002$ and $p < 0.0001$, respectively), hence suggesting an association between inhibition of ETNK1 activity and increased mitochondrial polarization and mass. The increased mitochondrial

polarization was further confirmed by using MitoProbe JC-1 and the mitochondrial membrane potential disrupter CCCP (Supplementary Fig. 3; 1.27 and 1.28 fold increase in ETNK1-N244S and ETNK1-KO lines compared to ETNK1-WT; $p = 0.0178$ and $p = 0.013$, respectively).

Transmission electron microscopy supported these findings, showing that in WT cells mitochondria are characterized by a round-shaped morphology, whereas in ETNK1-N244S and ETNK1-KO cells they deviate from circularity (measured circularity on segmented mitochondria: $0.82 ± 0.04$ in ETNK1-WT; $0.69 ± 0.05$ in ETNK1-N244S, and $0.66 ± 0.06$ in ETNK1-KO, $p < 0.0001$ with one-way ANOVA for both ETNK1-N244S and ETNK1-KO compared to ETNK1-WT). Moreover, in WT cells mitochondria hold higher electron density compared to both ETNK1-N244S and ETNK1-KO. Indeed, mutated lines showed a bigger and polymorphic shape with regions of low electron density in which the mitochondria crests were clearly defined, thus suggesting an increased activity[12,13] (Fig. 1c–f; 1.72 and 2.02 fold increase in ETNK1-N244S and ETNK1-KO compared to ETNK1-WT; $p < 0.0001$ for both N244S and KO compared to WT).

As glycolysis, the tricarboxylic acid cycle (TCA), and mitochondrial respiration are tightly coupled processes, we sought to investigate if the alteration in mitochondrial activity was associated with changes in the glycolytic or TCA cycle profiles in ETNK1-N244S cells. Whole-transcriptome sequencing was used to analyze the expression level of all glycolytic, pentose phosphate pathway (PPP), and TCA enzymes, while targeted mass spectrometric metabolite profiling and nuclear magnetic resonance (NMR) were applied to quantify the intracellular metabolite levels of these pathways as well as glucose consumption and lactate excretion, respectively (Supplementary Fig. 4; Supplementary Data 2). These analyses did not reveal significant intracellular changes at either gene or metabolite level, with one marked exception: the intracellular levels of lactate were significantly lower in both mutated and KO cells (1.69 and 1.84 fold decrease in ETNK1-N244S and ETNK1-KO lines compared to ETNK1-WT; both $p < 0.0001$). This was accompanied by a small but significant decrease in lactate excretion and glucose consumption ($p < 0.01$) in ETNK1-N244S cells. These data suggest that glycolytic function and TCA cycle are not globally altered in the ETNK1-driven leukemogenesis. Instead, the observed, focal decrease in lactate concentration corroborates the hypothesis of increased mitochondrial activity and of pyruvate import into the TCA cycle at the expense of lactate production.

Given the importance of mitochondrial PE for the oxidative phosphorylation pathway[10,11], we initially hypothesized that the alteration of mitochondria membrane potential in the presence of mutated or knock-out ETNK1 could be due to an alteration in the mitochondrial membrane composition and to a reduced PE fraction. To test this hypothesis we generated a whole-lipidome mass spectrometry profile of purified mitochondria as well as cell membranes in 293 ETNK1-WT and genome-edited ETNK1-N244S cells. Globally, we analyzed a total of 21 lipid families (Supplementary Fig. 5A), comprising 217 different PE species characterized by different fatty acid lengths and unsaturated-bond numbers. Differential lipidome profile analysis failed to reveal significant changes in the absolute or relative cell-membrane and mitochondria PE concentration (Supplementary Figs. 5–7) as well as in the concentration of related phospholipids, such as PC and phosphatidylserine (PS), therefore suggesting the presence of mechanisms able to overcome the decreased intracellular P-Et concentration. Similarly, no difference was detected in fatty acid composition (Supplementary Fig. 5B) or in PE double-bond number and fatty acid length (Supplementary

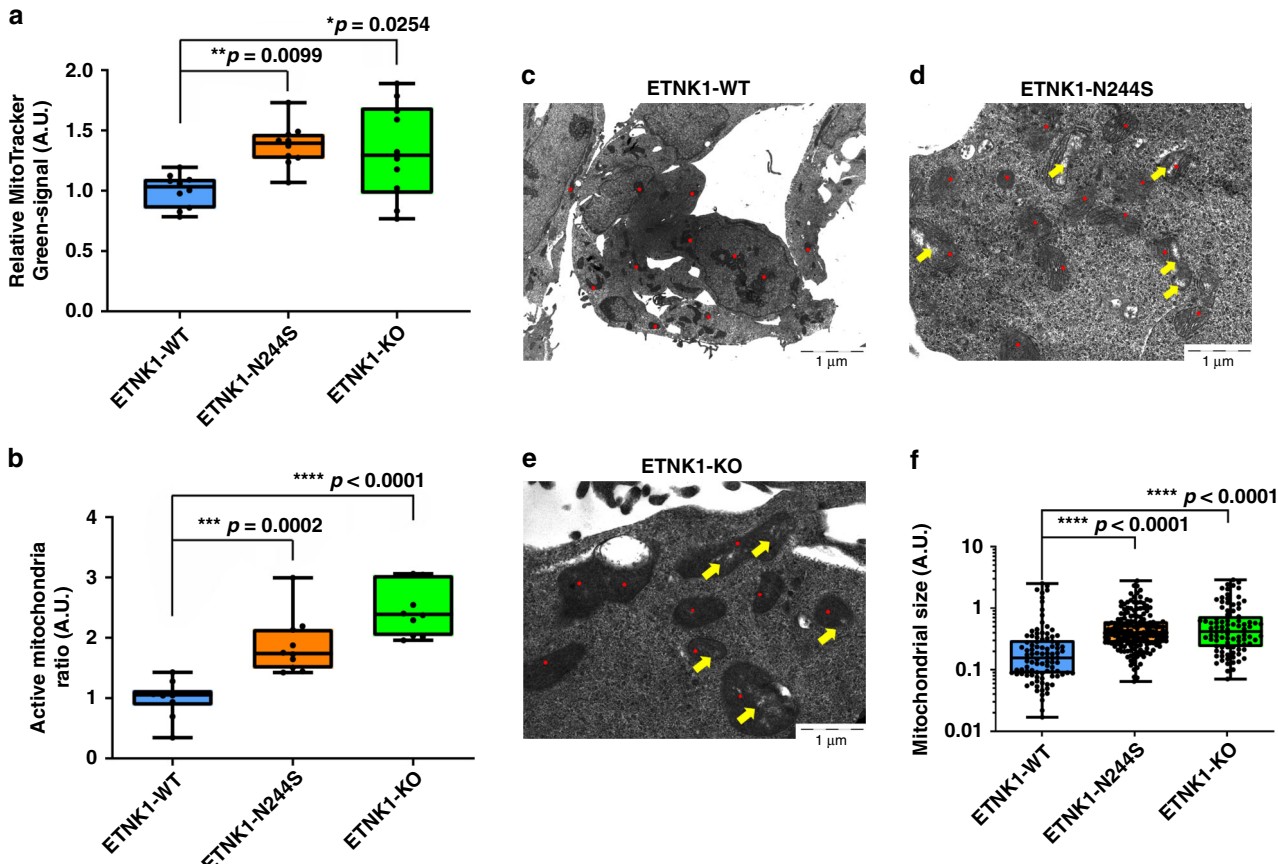

**Fig. 1 Mitochondria morphology and activity. a** Boxplots representing the MitoTracker Green analysis of ETNK1-WT, N244S, and KO CRISPR cell lines. Values represent relative units normalized on the ETNK1-WT signal. Statistical analyses were performed using one-way ANOVA with Tukey's post-hoc test. The boxplots delimit the interquartile range; the central bar represents the median; the whiskers extend from minimum to maximum ($n = 10$ representative fields). At least 200 cells were analyzed. **b** Boxplots showing the MitoTracker Red-to-Green signal ratio as assessed on ETNK1-WT, N244S, and KO CRISPR cell lines normalized on ETNK1-WT. Statistical analyses were performed using one-way ANOVA with Tukey's post-hoc test. The boxplots delimit the interquartile range; the central bar represents the median; the whiskers extend from minimum to maximum ($n = 10$ representative fields). At least 200 cells were analyzed. **c–e** Electron microscopy scans relative to ETNK1-WT **c**, N244S **d**, and KO **e** CRISPR cell lines. Red dots highlight the position of individual mitochondria. Yellow arrows point to low electron density areas. A total of 13 individual cells were analyzed. **f** Quantification of mitochondria size in ETNK1-WT, N244S, and KO CRISPR cell lines as assessed by electron microscopy. Statistical analyses were performed using one-way ANOVA with Tukey's post-hoc test. The boxplots delimit the interquartile range; the central bar represents the median; the whiskers extend from minimum to maximum (ETNK1-WT: $n = 96$ representative fields; ETNK1-N244S: $n = 172$ representative fields; ETNK1-KO: $n = 90$ representative fields; the experiment was performed once). Source data are provided as a Source data file.

Fig. 6). To further validate our findings, we conducted lipidome mass spectrometry analysis on Ficoll-purified primary bone marrow mononuclear cells of aCML patients, either positive (3) or negative (3) for *ETNK1* mutations, evaluating 10 of the most important lipid classes. The results indicated no differences in both the total amount and the composition of lipids in our patients (Supplementary Fig. 8A, B), confirming our previous findings. Decreased enzymatic activities are often compensated by the upregulation of alternative pathways. Whole-transcriptome differential expression analysis between ETNK1-WT and ETNK1-N244S lines revealed the presence of only 119 differentially expressed genes (FDR < 0.1; Supplementary Data 3), suggesting a very limited role of ETNK1 variants in modulating gene expression. Of them, 104 were upregulated and 15 downregulated. None of the differentially expressed genes were ascribable to ontologies related to lipid biosynthesis.

In line with these findings, the analysis of cell membrane rigidity by means of atomic-force indentation assays (Supplementary Fig. 9A–D) failed to reveal substantial differences among ETNK1-WT, ETNK1-N244S, and ETNK1-KO cells. Taken globally, these data indicated that human cells are able to

synthesize normal concentration of PE even in a condition of low intracellular P-Et, therefore ruling out a critical role for cell membrane PE in the oncogenesis mediated by *ETNK1* mutations.

**P-Et restores a normal mitochondrial activity in the presence of mutated ETNK1.** Recently, Gohil and colleagues demonstrated that treatment with meclizine, a known inhibitor of phosphoethanolamine cytidylyltransferase 2 (PCYT2), the second step in the Kennedy pathway downstream to ETNK1, leads to a potent inhibition of mitochondria respiration[14] and accumulation of P-Et. Similarly, Ferreira and colleagues showed that supplementation of MCF-7 cells with millimolar concentration of P-Et caused an abrupt decrease in mitochondria membrane potential[15]. Owing to these preliminary evidences, we turned our attention to P-Et, investigating a potential role for P-Et (or lack thereof) in the regulation of mitochondria activity. In our previous work, we showed that *ETNK1* somatic mutations impair ETNK1 catalytic activity, therefore resulting in a decrease in P-Et intracellular concentration[6]. Hence, we hypothesized that the decreased P-Et concentration in ETNK1-mutated and knock-out

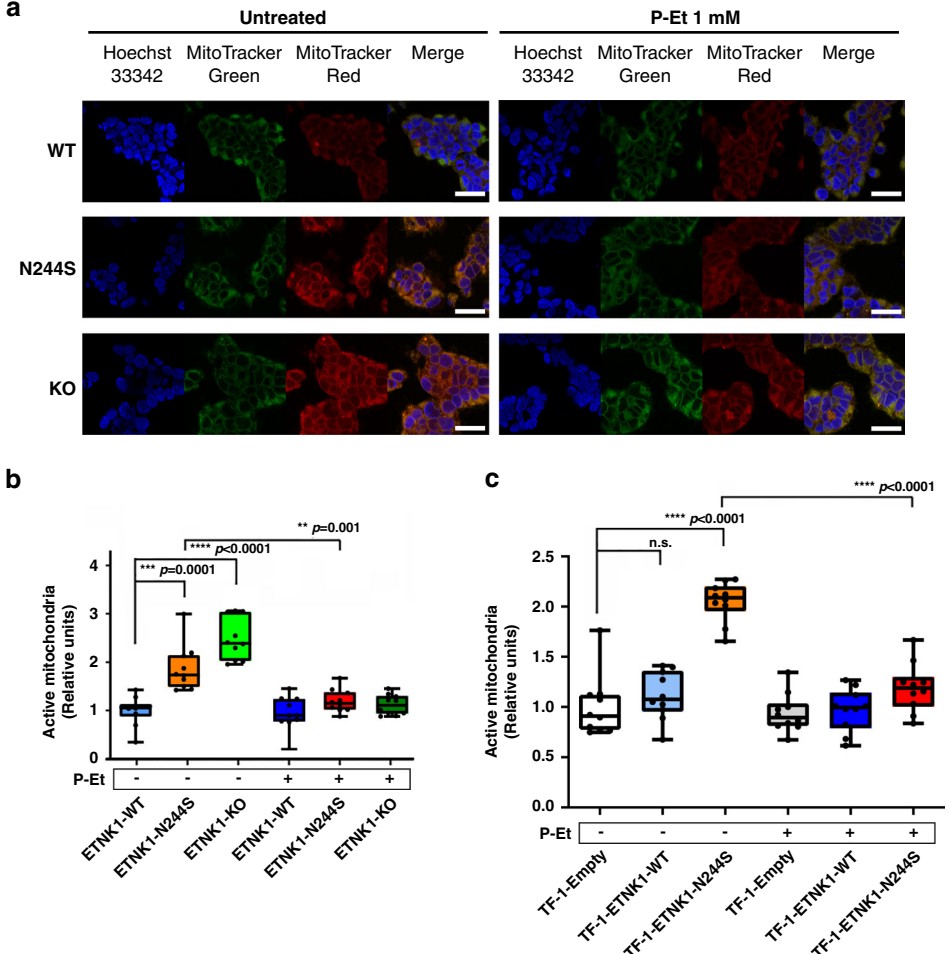

**Fig. 2 Mitochondria activity in the absence and presence of P-Et. a** Confocal microscopy of MitoTracker Red/Green signal in ETNK1-WT, N244S, and KO CRISPR cell lines in the absence (left) and presence (right) of 1 mM P-Et for 24 h. At least 200 cells were analyzed. The scale bar corresponds to 40 μm. **b** Boxplots showing the MitoTracker Red-to-Green signal ratio of ETNK1-WT, N244S, and KO CRISPR cell lines normalized on ETNK1-WT in the absence and in presence of 1 mM P-Et for 24 h. The boxplots delimit the interquartile range; the central bar represents the median; the whiskers extend from minimum to maximum ($n = 10$ representative fields). Statistical analyses were performed using one-way ANOVA with Tukey's post-hoc test. **c** MitoTracker Red-to-Green signal in the absence and in presence of 1 mM P-Et (for 24 h) in the myeloid TF-1 model. Statistical analyses were performed using one-way ANOVA with Tukey's post-hoc test. The boxplots delimit the interquartile range; the central bar represents the median; the whiskers extend from minimum to maximum ($n = 10$ representative fields). At least 200 cells were analyzed. Source data are provided as a Source data file.

cells, confirmed by liquid chromatography tandem–mass spectrometry (LC–MS/MS) (Supplementary Fig. 10; 3.42 and 3.24 fold decrease in ETNK1-N244S and ETNK1-KO lines compared to ETNK1-WT; $p < 0.0001$), could be responsible for the increased mitochondrial activity. To test this hypothesis, we repeated MitoTracker Red/Green experiments in presence/absence of P-Et (1 mM). ETNK1-N244S and ETNK1-KO cells treated with 1 mM P-Et showed a complete restoration of the normal membrane potential (Fig. 2a, b; 1.56 and 2.21 fold decrease in ETNK1-N244S and ETNK1-KO compared to untreated; $p = 0.0001$ and $p < 0.0001$, respectively). Conversely, no effect was detected in ETNK1-WT, where treatment with P-Et did not alter the basal mitochondrial activity level (Fig. 2a, b; 1.06 fold decrease; $p > 0.05$). As the 293 is not a hematological line and ETNK1 was found mutated in hematological neoplasms, we sought to replicate our data in the myeloid TF-1 model (Supplementary Fig. 11). In line with previous findings, MitoTracker analysis showed a strong increase in mitochondrial activity (Fig. 2c; 1.84 fold increase; $p < 0.0001$) in TF-1-ETNK1-N244S line compared to TF-1-ETNK1-WT. Treatment of myeloid TF-1 cells with P-Et

was able to fully revert the phenotype (Fig. 2c; 1.72 fold decrease compared to untreated; $p < 0.0001$).

As mutation H243Y was also reported in previous studies[6,7], we sought to investigate the functional effect of this second mutation. In line with data obtained with the N244S mutation, both MitoTracker and MitoProbe JC-1 analyses performed on myeloid TF-1 cells overexpressing ETNK1 mutant H243Y (TF-1-ETNK1-H243Y) showed an increase in mitochondrial activity when compared to TF-1-Empty cells (Supplementary Fig. 12; 1.48 fold increase; $p = 0.0003$; Supplementary Fig. 13; 1.41 and 1.36 fold increase in TF-1-ETNK1-H243Y and TF-1-ETNK1-N244S lines compared to TF-1-Empty; $p = 0.0021$ and $p = 0.0084$, respectively). These analyses confirmed also a phenotype reversion after the treatment with P-Et (Supplementary Fig. 12; 1.42 fold decrease compared to untreated; $p = 0.0014$). Taken globally, these data suggested a direct role for P-Et down-regulation in modulating mitochondria membrane potential.

**ETNK1-mutated lines show an increased cell respiration rate.** To assess if the apparent increase in mitochondria activity in

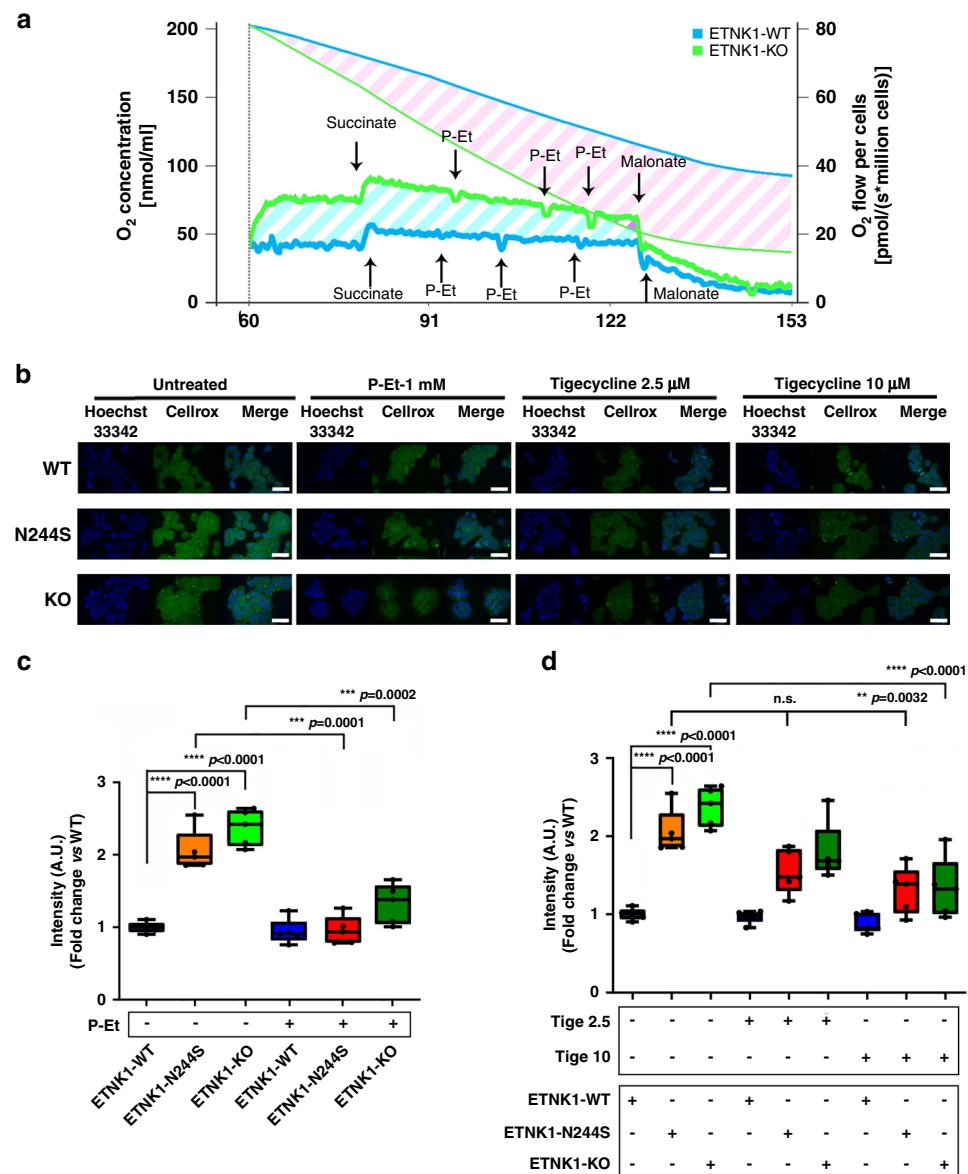

**Fig. 3 Mitochondria respiration and ROS production. a** Representative graph of $O_2$ consumption rate (pmol/(s*10$^6$ cells)) and $O_2$ concentration (nmol/ml) in ETNK1-WT and KO CRISPR lines ($n = 4$), measured by high-resolution respirometry Oroboros O2K (1 million cells). The thick blue and green lines represent the oxygen consumption rate of ETNK1-WT and KO cells, respectively; the thin blue and green lines show the oxygen concentration in ETNK1-WT and KO oxygraph chambers, respectively. The diagonal cyan and pink stripes highlight, respectively, the difference in oxygen consumption rate and oxygen concentration of ETNK1-WT and KO cells. **b** Intracellular reactive oxygen species as assessed by confocal microscopy using the CellROX reagent. ROS analysis was performed in ETNK1-WT, N244S, and KO CRISPR lines in the absence and presence of P-Et 1 mM, tigecycline 2.5 μM and tigecycline 10 μM. Both the P-Et and the tigecycline treatments lasted 24 h. At least 200 cells were analyzed. The scale bar corresponds to 40 μm. **c** ROS quantification in ETNK1-WT, N244S, and KO CRISPR lines in the absence and presence of 1 mM P-Et. The boxplots delimit the interquartile range; the central bar represents the median; the whiskers extend from minimum to maximum ($n = 5$ representative fields). Statistical analyses were performed using one-way ANOVA with Tukey's post-hoc test. **d** ROS quantification in ETNK1-WT, N244S, and KO CRISPR lines in the absence and presence of 2.5 and 10 μM tigecycline. The boxplots delimit the interquartile range; the central bar represents the median; the whiskers extend from minimum to maximum ($n = 5$ representative fields). Statistical analyses were performed using one-way ANOVA with Tukey's post-hoc test. Source data are provided as a Source data file.

ETNK1-mutated lines was accompanied by an increased cell respiration, we measured the oxygen consumption rate (reported as $O_2$ pmol/(s*10$^6$ cells)) and concentration (nmol/ml) in 293 ETNK1-WT and KO lines (Fig. 3a). In accordance with Mito-Tracker experiments showing an increased mitochondrial respiratory activity in mutated lines, the oxygen consumption rate resulted strongly increased in ETNK1-KO lines compared to WT (Fig. 3a). Similarly, the oxygen concentration curve showed a much faster decrease in ETNK1-KO than in the WT line over the entire course of the experiment, hence indicating a marked

increase in the cellular respiration rate in the former cells (Fig. 3a). Treatment with 100 μM, 1, and 2 mM P-Et (Fig. 3a: first, second, and third P-Et arrow) caused in all cases a rapid decrease of the respiration rate, therefore suggesting a role for P-Et as a negative regulator of mitochondria respiration.

**ETNK1 mutations are responsible for increased mitochondrial ROS production, which can be reverted back to a normal level by treatment with P-Et or tigecycline.** A large fraction of the total intracellular ROS is generated during the process of

oxidative phosphorylation in the inner mitochondrial membrane. Hence, oxidative phosphorylation and generation of ROS are tightly connected processes. To investigate if the increased mitochondrial membrane potential in CRISPR ETNK1-mutated lines could be also associated with an abnormal production of ROS, we assessed their intracellular level by using the CellROX Green Reagent fluorogenic dye. CellROX dye is weakly fluorescent in a reduced state, whereas it exhibits a bright green fluorescence upon oxidation by ROS, therefore acting as an effective probe to estimate the levels of oxidative stress in live cells. As expected, we detected a significant increase in ROS production in mutated and knock-out ETNK1 lines compared to the WT ones (Fig. 3b, c; 2.05 and 2.38 fold increase; $p < 0.0001$). Similar results were also achieved using the myeloid TF-1 model (Supplementary Fig. 14; 1.74 and 2.04 fold increase in TF-1-ETNK1-H243Y and TF-1-ETNK1-H244S lines, respectively; $p < 0.0001$). In order to validate these initial results, we repeated the same experiments after the supplementation of the ROS quencher N-acetyl-L-cysteine and after the exposure to UV rays to increase cellular stress and ROS production. As expected, treatment with N-acetyl-L-cysteine completely abolished the increased CellROX signal in both ETNK1-N244S and ETNK1-KO lines (Supplementary Fig. 15; 1.38, 2.42, and 2.83 fold decrease in treated ETNK1-WT, ETNK1-N244S, and ETNK1-KO lines compared to untreated; $p = 0.0002$ in WT and $p < 0.0001$ in N244S and KO lines). Conversely, UV-irradiation led to an increased ROS production in all the lines, with a more pronounced increment in the WT, where the initial ROS levels were low (Supplementary Fig. 16; 1.98, 1.72, and 1.41 fold increase in UV-irradiated ETNK1-WT, ETNK1-N244S, and ETNK1-KO lines compared to unexposed; $p < 0.0001$). To assess if the production of ROS was modulated by the intracellular P-Et content, we re-evaluated ROS levels after supplementation of target cells with 1 mM P-Et. Notably, treatment with P-Et restored ROS production to near normal levels in both CRISPR lines (Fig. 3b, c; 2.14 and 1.79 fold decrease in treated ETNK1-N244S and ETNK1-KO lines compared to untreated; $p = 0.0001$ and $p = 0.0002$, respectively) and TF-1 (Supplementary Fig. 14; 1.51 and 1.57 fold decrease in treated TF-1-ETNK1-H243Y and TF-1-ETNK1-H244S lines compared to untreated; $p < 0.0001$), therefore highlighting a role for P-Et as a potent regulator of mitochondria activity and generation of ROS.

Given the direct connection of mitochondrial oxidative phosphorylation activity and ROS production, we hypothesized that the synthesis of ROS could be controlled by reducing the activity of mitochondria machinery. Tigecycline is an antibiotic used to treat a number of bacterial infections[16]. Its mechanism of action relies on the inhibition of bacterial protein synthesis. Because of the similarity of mitochondrial and bacterial ribosomes, tigecycline also inhibits the synthesis of mitochondrial proteins required for the oxidative phosphorylation machinery[17]. To assess if tigecycline could dampen mitochondrial ROS production in the context of ETNK1-mutated cells, we treated CRISPR ETNK1 lines with 2.5 and 10 μM tigecycline for 24 h and ROS production was assessed by mean of CellROX assay (Fig. 3b, d). In line with the known ability of tigecycline to modulate mitochondria oxidative phosphorylation, treatment in ETNK1-N244S and KO lines reduced ROS levels to an intermediary level (Fig. 3b, d; 1.57 and 1.78 fold decrease in 10 μM tigecycline-treated ETNK1-N244S and ETNK1-KO lines compared to untreated; $p = 0.0032$ and $p < 0.0001$, respectively), while virtually no effect could be detected in WT controls (Fig. 3b, d; 1.03 and 1.13 fold decrease in ETNK1-WT treated with 2.5 and 10 μM tigecycline; $p > 0.05$).

**ETNK1 mutations cause accumulation of 8-Oxoguanine DNA lesions and induction of a mutator phenotype that can be reverted with P-Et or tigecycline treatment.** The role of ROS as

mutagenic agents is well-documented as they are known to play a major role in the onset and clonal evolution of many cancers[18]. Hundreds of different oxidative DNA modifications have already been identified in eukaryotic genomes; in particular, guanine is extremely vulnerable to ROS damage, due to its low redox potential[19]. The main product of ROS-mediated guanine oxidation is the 7,8-dihydro-8-oxo-2′-deoxyguanosine (oxoG; Fig. 4a). If not repaired by the physiologic DNA repair mechanisms, i.e. by the oxoG-glycosidases, oxoG can pair with adenine, causing $G \rightarrow T$ transversions (Fig. 4b). Hence, oxoG can be used as a biomarker of ROS-mediated genomic DNA damage. To leverage this information and to assess if ETNK1 mutations could cause accumulation of DNA lesions, we generated ChIP-Seq data for the CRISPR ETNK1-N244S line using an antibody raised against the oxoG and we compared the oxoG signal with the WT one. The analysis of the oxoG signal at whole-genome level using custom bioinformatics tools (details can be found in the "Methods" section) revealed a significant increase in oxoG in ETNK1-N244S (Fig. 4c; $p = 0.018$; Wilcoxon signed-rank test). As the accumulation of intracellular ROS was accompanied by a significant increase of oxoG lesions in the nucleus of ETNK1-mutated cells, we sought to investigate if these lesions were driving the onset of a mutator phenotype, therefore overcoming the physiologic DNA repair mechanisms of the target cells. To test this hypothesis, we applied the 6-thioguanine (6-TG) resistance assays to our CRISPR cell models. In the 6-TG assay, cells are grown in semisolid medium in the presence of 6-TG, which, following phosphorylation by the hypoxanthine phosphoribosyl transferase 1 (HPRT1), is incorporated into the nascent gDNA, causing cell death. The only cells surviving the treatment are those that incorporated inactivating HPRT1 mutations. Therefore, the number of surviving cells can be used to estimate the relative mutation rate of each sample. In line with ChIP-Seq data, 6-TG assays demonstrated a significant increase in the number of resistant colonies in mutated and knock-out ETNK1 lines when compared to the reference (Fig. 4d; Supplementary Fig. 17; Supplementary Data 4; 5.4 and 4.9 fold increase compared to the WT line in ETNK1-N244S and KO, respectively; $p < 0.0001$). Also, when performed in CRISPR cells pre-treated with 1 mM P-Et or 2.5 μM tigecycline, 6-TG assay caused an almost complete reversion of the mutant phenotype back to the normal level in mutated and knock-out lines (Fig. 4d; Supplementary Data 5; P-Et treatment: 5.74 and 4.14 fold decrease in pre-treated ETNK1-N244S and ETNK1-KO lines compared to untreated, $p < 0.0001$; tigecycline treatment: 2.17 and 3.45 fold decrease in pre-treated ETNK1-N244S and ETNK1-KO lines compared to untreated, $p < 0.0001$), indicating that the presence of an ETNK1-mediated mutator phenotype is fully dependent on mitochondria increased activity, and can be controlled by physiological or pharmacological modulators of oxidative phosphorylation.

**Mutant ETNK1 induces DNA double-strand breaks (DSBs).** Under normal conditions, human DNA is subjected to a constant process of oxidative damage mediated by ROS that are generated in large part as a byproduct of mitochondria oxidative phosphorylation[20]. It is estimated that in a single cell cycle almost 5000 single-stranded DNA breaks can occur as a result of ROS production. Approximately 1% of these DNA breaks is converted into DSBs, mainly during DNA replication, while the remaining 99% is repaired[20,21]. As ETNK1+ clonal disorders, such as aCML, are typically characterized by chromosome instability and clonal evolution[22], we sought to investigate if the ROS-mediated genotoxic insult operating in ETNK1-mutated lines could be also associated with an increase in DNA DSBs. To achieve this goal, we analyzed the level of phosphorylated histone H2AX (γH2AX),

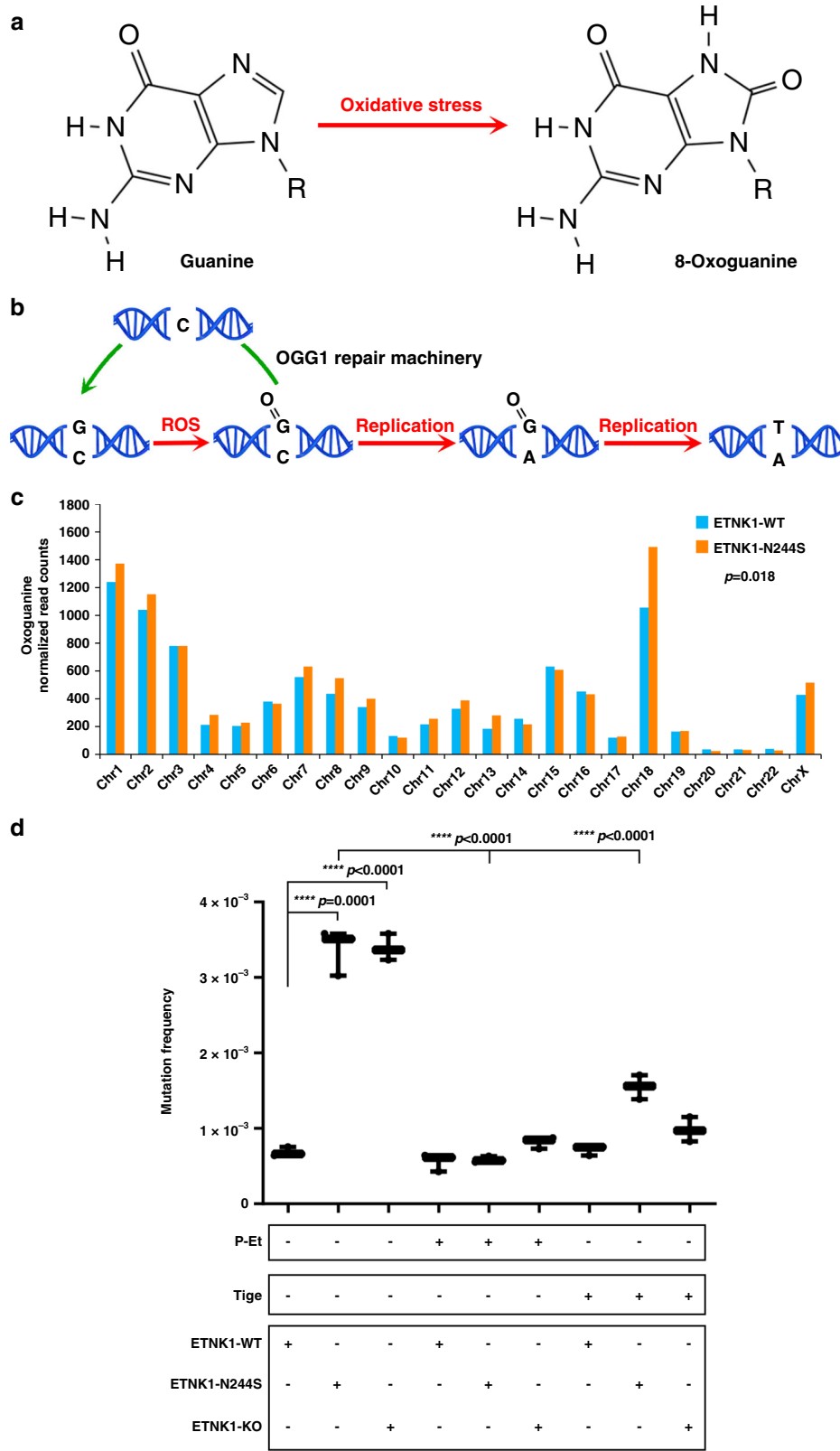

a well-known marker of DNA DSBs[23], by confocal microscopy. Comparison of ETNK1-N244S and ETNK1-WT lines revealed a sharp increase in the number of γH2AX foci (Fig. 5a, c; 2.52 fold increase; $p = 0.0002$) in the former. Similar results were achieved in the presence of ETNK1-KO (Fig. 5a–c; 2.51 fold increase; $p = 0.0002$), indicating that the suppression of ETNK1 activity and the ensuing decrease in P-Et intracellular concentration are

followed by an increase in DNA DSBs. To further confirm the association between ROS and DNA DSBs, we performed the same experiment after the supplementation of N-acetyl-L-cysteine. Treatment with the ROS quencher led to a strong decrease in the γH2AX signal, therefore confirming the central role of ROS production in the generation of the DSB DNA damage (Supplementary Fig. 18; 2.11 and 1.88 fold decrease in treated ETNK1-

**Fig. 4 Oxoguanine analysis and 6-thioguanine resistance. a** Diagram showing the chemical reaction responsible for the generation of oxoG from G after exposure of gDNA to reactive oxygen species. **b** Scheme of the oxoG-mediated DNA damage. In the presence of the modified base, the gDNA may undergo two different destinies: either the base is excised, owing to the recruitment of the oxoG DNA glycosylase 1 (OGG1) repair machinery before the onset of a new replication cycle, or it causes the misincorporation of an adenine in the complementary strand, eventually leading to a G:C to T:A transversion. **c** Per-chromosome quantification of oxoG binned read counts in ETNK1-WT and N244S CRISPR lines following total read counts normalization; ($n = 2$). Statistical analysis was performed using a Wilcoxon matched-pairs signed rank test. **d** Mutation frequency assessed by 6-thioguanine assays in ETNK1-WT, N244S, and KO CRISPR lines in the absence or presence of 1 mM P-Et or 2.5 μM tigecycline (cell were pretreated for 15 days; 1 million of cells was plated and exposed to 30 μM of 6-TG for 15 days, while 1500 cells were plated for 15 days, as control). The boxplots delimit the interquartile range; the central bar represents the median; the whiskers extend from minimum to maximum ($n = 3$ independent experiments). Statistical analyses were performed using one-way ANOVA with Tukey's post-hoc correction. Source data are provided as a Source data file.

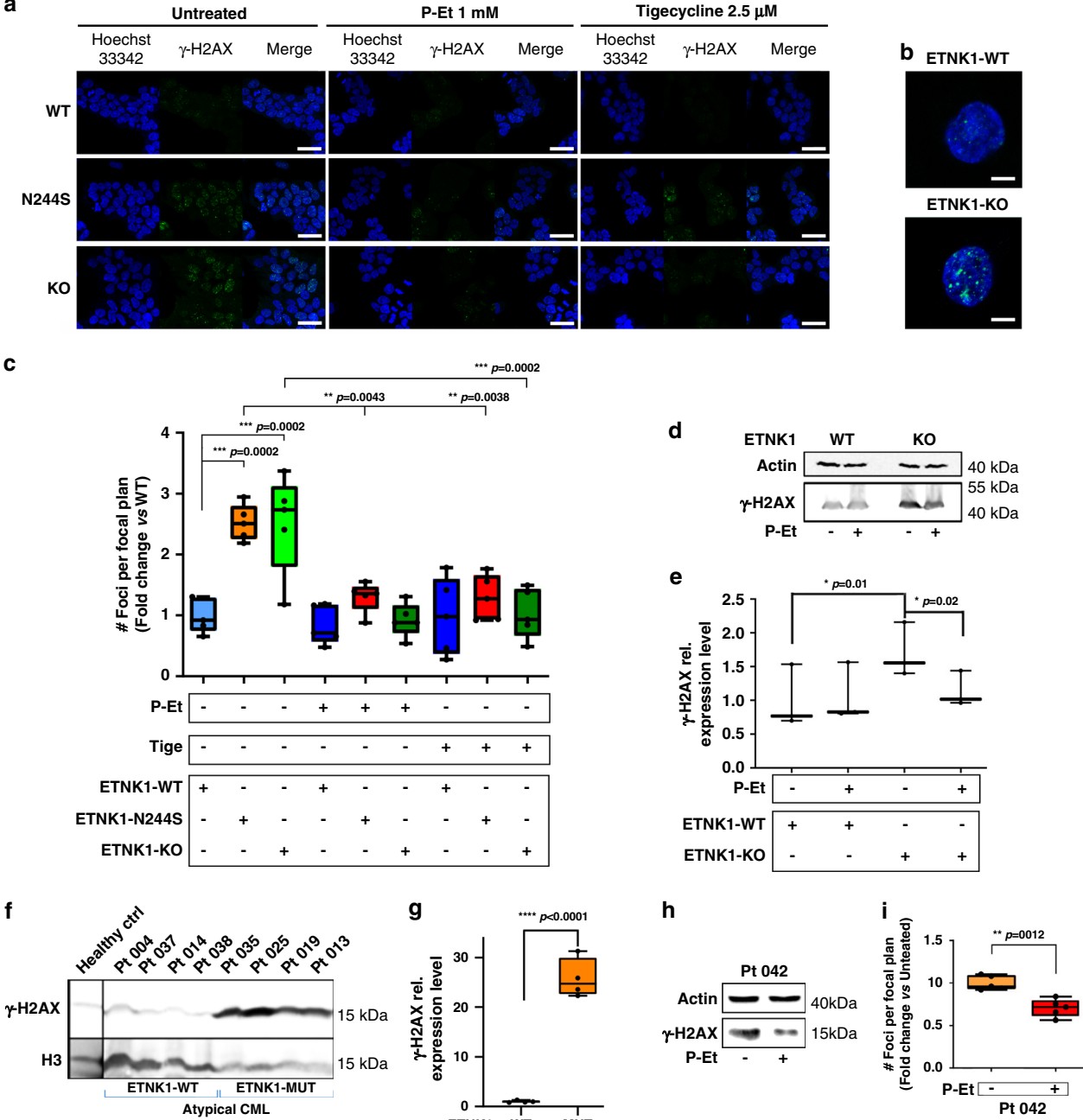

**Fig. 5 Double-strand DNA damage. a** Confocal microscopy analysis of γH2AX foci in ETNK1-WT, N244S, and KO CRISPR lines in the absence or presence of 1 mM P-Et or 2.5 μM tigecycline for 24 h. At least 200 cells were analyzed. The scale bar corresponds to 40 μm. **b** Detail of γH2AX foci in individual ETNK1-WT and ETNK1-KO CRISPR cells. A total of 200 cells were analyzed for ETNK1 and KO lines. The Z-stack was processed to generate a 3D cell image in a total of five cells per type. The scale bar corresponds to 8 μm. **c** γH2AX signal quantification in ETNK1-WT, N244S, and KO CRISPR lines in the absence or presence of 1 mM P-Et or 2.5 μM tigecycline for 24 h. The boxplots delimit the interquartile range; the central bar represents the median; the whiskers extend from minimum to maximum ($n = 5$ representative fields). Statistical analyses were performed using one-way ANOVA with Tukey's post-hoc test. **d** Anti-γH2AX western blot analysis on ETNK1-WT and ETNK1-KO CRISPR lysates in the absence and presence of 1 mM P-Et for 24 h. The analysis was performed in triplicate. Gel loading was normalized using actin. **e** Densitometric analysis of the western blot shown in panel **d**. Statistical analyses were performed using one-way ANOVA with Tukey's post-hoc test. The boxplots delimit the interquartile range; the central bar represents the median; the whiskers extend from minimum to maximum ($n = 3$ biologically independent replicates). **f** Anti-γH2AX western blot analysis on ETNK1-WT (left; 4 cases) and ETNK1-mutated (right; 4 cases) aCML patient samples. Given the limited amount of primary samples, this blot was performed only once. Gel loading was normalized using total H3. **g** Densitometric analysis of the western blot shown in panel **f** ($p < 0.0001$). The signal represents corresponds to the mean, H3-normalized anti-γH2AX signal of ETNK1-WT and ETNK1-mutated aCML patients, respectively. The boxplots delimit the interquartile range; the central bar represents the median; the whiskers extend from minimum to maximum ($n = 4$ biologically independent samples). Statistical analysis was performed using a two-sided $t$-test. **h** Anti-γH2AX Western blot analysis on cell lysates from an ETNK1-mutated aCML bone marrow sample treated/non-treated with 1 mM P-Et for 24 h. Given the limited amount of primary samples, this blot was performed only once. **i** γH2AX signal quantification in bone marrow primary cells obtained from patient Pt 042 (ETNK1-N244S+) in the absence or presence of 1 mM P-Et. At least 200 cells were analyzed. The boxplots delimit the interquartile range; the central bar represents the median; the whiskers extend from minimum to maximum ($n = 5$ representative fields). Statistical analyses were performed using a two-sided $t$-test. Source data are provided as a Source data file.

N244S and ETNK1-KO lines compared to untreated; $p < 0.0001$). Conversely, exposition of CRISPR cells to UV rays led to an increase in the intensity of the γH2AX signal in all cells, with a more pronounced effect in the ETNK1-WT line (Supplementary Fig. 19; 3.61, 1.64, and 1.56 fold increase in exposed ETNK1-WT, ETNK1-N244S, and ETNK1-KO lines compared to unexposed; $p < 0.0001$, $p = 0.015$, and $p = 0.036$, respectively). To search for a causative role of the ETNK1-dependent mitochondrial machinery in the onset of the chromosome instability phenotype, we repeated the same experiments in the presence of 1 mM P-Et or 2.5 μM tigecycline. Both treatments were able to restore the rate of DNA DSBs back to a normal level (Fig. 5a, c; 1.94 and 2.71 fold decrease in P-Et-treated ETNK1-N244S and ETNK1-KO lines compared to the untreated; $p = 0.0043$ and $p < 0.0001$, respectively; 1.95 and 2.45 fold decrease in tigecycline-treated ETNK1-N244S and ETNK1-KO lines compared to the untreated; $p = 0.0038$ and $p = 0.0002$, respectively), therefore supporting the hypothesis of a direct involvement of the ETNK1–P-Et–Mitochondria axis in the induction of DNA damage. Anti-γH2AX western blots on ETNK1-WT and ETNK1-KO lysates confirmed the same finding (Fig. 5d, e). In line with data generated with the CRISPR/Cas9 N244S and KO models, over-expression of H243Y or N244S in the hematological TF-1 line revealed the same behavior, with cells overexpressing H243Y or N244S showing a strong increase in the number of γH2AX foci as compared with the control counterpart (Supplementary Fig. 20; 2.49 and 3.12 fold increase in TF-1-ETNK1-H243Y and TF-1-ETNK1-H244S, respectively; $p < 0.0001$). As expected, treatment with P-Et restored the normal phenotype (Supplementary Fig. 20; 1.66 and 1.78 fold decrease in treated TF-1-ETNK1-H243Y and TF-1-ETNK1-H244S lines compared to untreated; $p < 0.0001$).

To assess if the presence of ETNK1 somatic mutations could similarly associate with increased DNA damage in patients' samples, we analyzed by anti-γH2AX western blots Ficoll-purified primary bone marrow mononuclear cell lysates of aCML patients, either positive (4) or negative (4) for ETNK1 mutations, whose mutational landscape had been previously characterized by mean of paired exome-sequencing (Supplementary Data 6–13). In line with the results generated in CRISPR/Cas9 cell-line models, the ex vivo analysis done on primary aCML cells confirmed a strong activation of the γH2AX signal, which was restricted to the entire group of the ETNK1-mutated samples (Fig. 5f, g; 25.78 fold increase; $p < 0.0001$) and virtually absent in the remaining samples. In an ETNK1 + aCML patient whom bone marrow cells

were available (Supplementary Data 14), we tested γH2AX signal by western blot (Fig. 5h) and confocal microscopy (Fig. 5i) in the absence and presence of 1 mM P-Et. As shown for ETNK1+ cell lines, ex vivo experiments confirmed the reduction of γH2AX signal in the presence of the drug. In line with both CRISPR/Cas9 and retroviral models, also in patient's cells this phenomenon was accompanied by a decrease in ROS production (Supplementary Fig. 21; $p = 0.0002$).

**P-Et directly modulates mitochondrial complex II activity.** Although the data from mitochondria activity and ROS production were clearly pointing to a precise role for P-Et as a modulator of mitochondria activity, so far no known pathway was able to explain this connection. To dissect the precise mechanism by which P-Et intracellular levels were able to control mitochondria activity, we isolated the oxidative phosphorylation complexes I–IV[24] from both 293 ETNK1-WT and TF-1-Empty cells by immunoprecipitation from lysed cells (complexes I, II and IV) or from purified mitochondria (complex III). Subsequently, we measured the activity of each complex in absence/presence of increasing P-Et concentrations. This approach failed to show any effect of P-Et on complexes I, III, and IV (Fig. 6a, c, d; Supplementary Fig. 22A, C, D), but revealed a profound, dose-dependent decrease in redox activity of the mitochondrial complex II, both in the 293 line (Fig. 6b; P-Et 10 μM: 1.80 fold decrease; $p = 0.0012$; P-Et 20 μM: 7.40 fold decrease; $p < 0.0001$; P-Et 50 μM: 28.85 fold decrease; $p < 0.0001$) and in myeloid TF-1 cells (Supplementary Fig. 22B; P-Et 10 μM: 1.29 fold decrease; $p = 0.0004$; P-Et 20 μM: 1.92 fold decrease; $p < 0.0001$; P-Et 50 μM: 8.94 fold decrease; $p < 0.0001$), hence indicating that P-Et controls mitochondria potential through direct inhibition of complex II.

Mitochondrial complex II, also known as SDH, is both a central component of the mitochondria oxidative phosphorylation chain and of the TCA cycle, therefore representing a bridge between these two critical cellular processes. Together with complex I and complex III, complex II represents the top mitochondrial producer of ROS in terms of maximal capacity[25]. Although ROS generation through SDH is a complex and still not completely understood phenomenon, it is known that increased ROS production may occur in the presence of high intracellular succinate concentration, causing complex II to increment the rate of ubiquinone reduction and promoting reverse electron transfer to complex I[26]. In this scenario, we reasoned that P-Et could

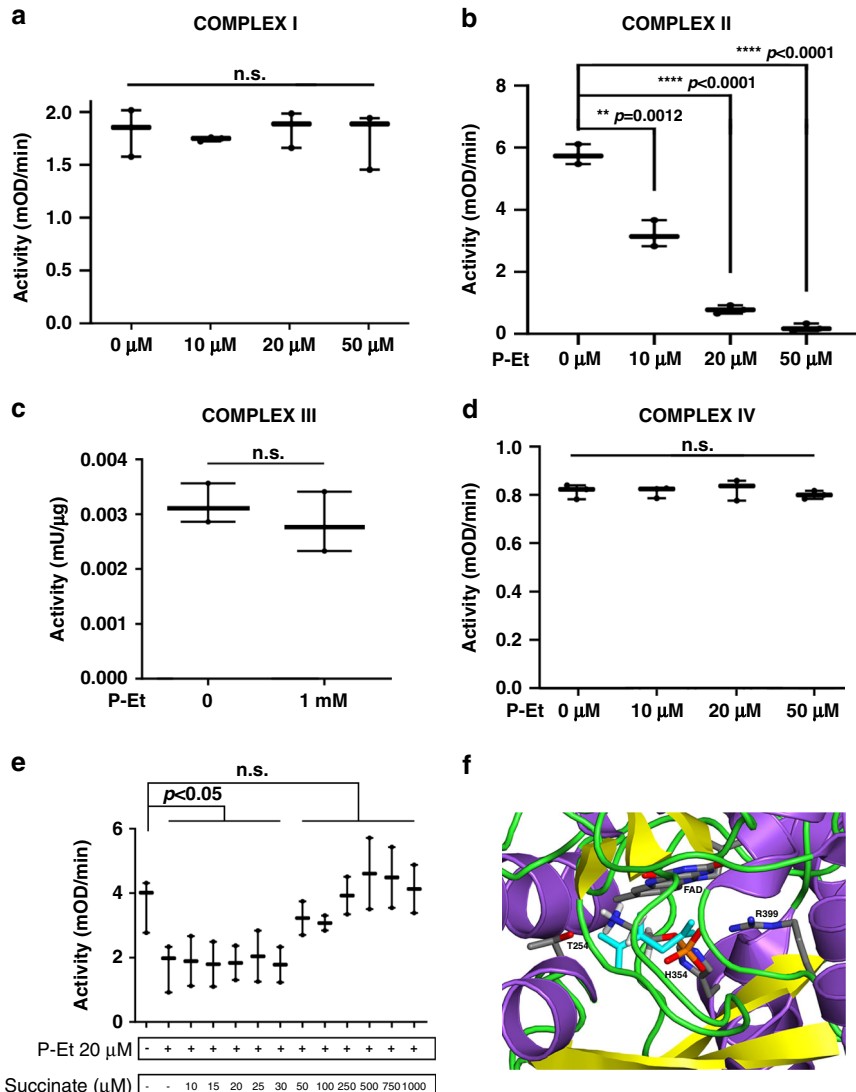

**Fig. 6 Activity of mitochondria complexes I–IV. a–d** Activity of ETNK1-WT mitochondria complexes I–IV in the presence of increasing concentrations of P-Et. The analysis of complexes I ($n = 3$ independent replicates), II ($n = 3$ independent replicates), and IV ($n = 3$ independent replicates) was performed on mitochondria lysates, while the analysis of complex III ($n = 3$ independent replicates) was performed on intact isolated mitochondria. The boxplots delimit the interquartile range; the central bar represents the median; the whiskers extend from minimum to maximum. Statistical analyses were performed using one-way ANOVA with Tukey's post-hoc test **a**, **b**, **d** and two-sided *t*-test **c**. **e** The activity of the mitochondria complex II was assessed in the absence and presence of 20 μM P-Et in combination with increasing concentrations of succinate. The boxplots delimit the interquartile range; the central bar represents the median; the whiskers extend from minimum to maximum ($n = 3$ independent replicates). Statistical analyses were performed using one-way ANOVA with Tukey's post-hoc test. **f** One of the best docking poses of P-Et onto the crystal structure of SDH (PDB: 1NEN) is shown. P-Et is represented in sticks colored by atom type. The protein is represented in cartoons colored according to secondary structure, i.e. helices, strands, and loops are, respectively, violet, yellow, and green. FAD and selected amino acid side chains in the catalytic site are represented in sticks colored by atom type. Succinate from the 1NEN complex is also shown in cyan sticks. Hydrogen atoms are shown only on P-Et. Source data are provided as a Source data file.

physiologically act like a brake, keeping ROS production from complex II at bay by negatively modulating SDH activity.

To gain insight into the specific mechanism by which P-Et could repress complex II, we analyzed its activity in competition assays in the presence of both P-Et and increasing concentration of succinate, the endogenous substrate of SDH. Treatment with 20 μM P-Et alone caused a significant decrease in complex II activity (Fig. 6b, e; Supplementary Fig. 22B, E); however, subsequent supplementation of succinate was able to restore the normal SDH activity starting from 50 μM (Fig. 6e) or 100 μM for TF-1 (Supplementary Fig. 22E), hence suggesting that P-Et acts as a competitive inhibitor of succinate for SDH activity.

In line with these data, automatic docking of P-Et and SDH suggested that P-Et can occupy the succinate binding site in an energetically favorable conformation, also mimicking succinate in some relevant interactions with SDH (Fig. 6f). In the predicted docking mode, the phosphate group of P-Et interacts with H354 and R399 like one of the two carboxylates of succinate, whereas the protonated amino-group interacts with FAD and approaches T254 similarly to the other carboxylate of succinate (Fig. 6f). Taken globally, these results strengthen our in vitro evidence that P-Et may act as a competitive inhibitor of SDH.

## Discussion

Owing to the introduction of Next Generation Sequencing technologies, the identification of somatic mutations present in human cancer cells has become relatively simple. However, the plain identification of these mutations has a very limited practical

impact if it is not accompanied by wet-lab studies dedicated to dissect the functional effects of each mutation. Despite their intrinsic complexity, these studies are invaluable not only to gain further insight into the specific prooncogenic mechanisms at work with each variant, but also to convert this information into effective treatments for patients.

*ETNK1* somatic mutations were identified for the first time by our group in aCML and CMML[6], by Lasho and colleagues in SM[7] and, more recently, in DLBCL by Zhou and colleagues[9]. In our first work, we showed that ETNK1 mutations caused a decreased ETNK1 enzymatic activity, hence translating into a reduced intracellular P-Et concentration. However, at that time the functional role of the mutated ETNK1 activity was completely unclear. In this work, we focused on the events occurring downstream of the impaired ETNK1 activity, trying to track-down the specific molecular mechanisms explaining why ETNK1 mutations are oncogenic.

As we know, cancer cells are in direct competition with each other for limited resources. At subclonal level, the existence of a strong selective pressure translates into the accumulation of new somatic mutations causing an increase in the overall fitness of the competing subclones. This process can occur through at least two different mechanisms: the first one is the acquisition of mutations directly improving the overall fitness. Variants like these translate into an immediate benefit for the mutated subclone, which in turn can, e.g. proliferate faster, have a more efficient metabolic profile, or can be protected from pro-apoptotic signals. The second mechanism does not affect the fitness of the target subclone directly; instead, it increases the chances for the target cell to incorporate new mutations through the induction of a mutator phenotype. While this mechanism does not lead to an immediate benefit in terms of increased fitness, it allows the cells bearing these mutations to accumulate new variants at an increased pace, hence, gaining momentum over time. Here, we showed that ETNK1 mutations, present in both the 293 line and in myeloid cells, operate through this second mechanism owing to their ability in modulating mitochondria activity, and ROS production. We also showed that the ETNK1 mutator phenotype can be generated both in CRISPR/Cas9 models as well as in cells over-expressing the mutated protein, hence suggesting that ETNK1 mutations likely operate through a dominant mechanism.

Taking into account the well-known ability of aCML to undergo clonal evolution towards a very aggressive, advanced phase, the risk of clonal evolution should be avoided at all costs. Although preliminary, our data suggest that treatment with P-Et, the metabolite whose production is reduced by ETNK1 mutations: (1) allows to quickly and efficiently decrease mitochondria hyperactivation; (2) normalizes ROS production; (3) reduces the accumulation of new mutations at levels comparable to ETNK1-WT cells, even though the exact mechanism by which this small, polar molecule is imported into the cells is at present unknown. Similarly, treatment with the antibiotic tigecycline, a drug that is routinely used to treat microbial infections, leads to very similar results. Tigecycline is a well-known inhibitor of mitochondria activity, owing to its ability in blocking mitochondria protein synthesis and to selectively induce cell death in several leukemia cell lines. The evidence that two molecules, P-Et and tigecycline, completely different in chemical structure but functionally converging at mitochondrial level, are indeed able to suppress the ETNK1-driven mutator phenotype strongly supports the notion that mitochondria are the center of this ETNK1-based prooncogenic machinery.

In conclusion, a new mechanism is shown here by which the reduced activity of mutant ETNK1 leads to the accumulation of new mutations through the reduced competition of P-Et with succinate, increased mitochondrial activity and ROS production

(Fig. 7). This mechanism can be blocked by either P-Et supplementation or tigecycline treatment, therefore, suppressing the accumulation of new mutations mediated by the ETNK1-dependent mutator phenotype.

Further studies will be required to assess if the treatment with tigecycline or P-Et is able to reduce the risk of progression of ETNK1+ leukemias.

## Methods

**Chemicals**. P-Et (Sigma) and 6-TG (Sigma) were dissolved in DMEM at 1 M and 1.5 mM concentration, respectively, aliquoted and stored at $-20\,^{\circ}C$. Succinate (Sigma) and tigecycline (Tigacil®; Pfizer) were dissolved in ddH₂O at 500 mM and 5 mM, respectively, aliquoted and stored at $-80\,^{\circ}C$.

**Cell lines**. Human embryonic kidney (HEK) cell line 293 was purchased from Thermo Fisher Scientific. Cells were cultured in DMEM (Lonza) medium supplemented with 10% FBS (Euroclone), 2 mM L-glutamine (Euroclone), 100 U/ml penicillin (Euroclone), 100 μg/ml streptomycin (Euroclone), 20 mM HEPES (Euroclone), and incubated in a humidified atmosphere at 37 °C with 5% $CO_2$. Cells were routinely screened for mycoplasma contamination (GATC Biotech AG). Cell number and viability were assessed by Trypan Blue (Sigma) count every 48 or 72 h.

ETNK1-WT, ETNK1-N244S, and ETNK1-KO clones were obtained as described in Ran et al.[27] with minor modifications, by cotransfecting 293 cells with pCas9_WT_GFP vector (derived from the Addgene vector pCas9D10A_GFP; please see Site-directed mutagenesis), pSG-U6-gRNA vector (GenScript) encoding for the following gRNA sequence: TATTCATGCACACAATGGC, and with a single-stranded DNA oligonucleotide (ssODN) purchased by IDT Integrated DNA Technology. The ssODN sequence is: TGATACTTAAGTTTTTCTTTTTATGCG GTTTTGTTTTAAACAGGCTAATAGCTCGTCAGCTTGCTAAAATCCATGCT ATTCATGCACACAGTGGCTGGATCCCCAAATCTAATCTTTGGCTAAAGA TGAAAGTATTTCTCTCTCATTCCCACAGGATTTGCAGATGAAGACATTAA TAA

Briefly, 240,000 cells were resuspended in 24 μl transfection solution supplied in the Amaxa SF cell line 4D-Nucleofector X kit S (Lonza), and transferred in a tube containing 250 ng pCas9_WT_GFP plasmid, 250 ng pSG-U6-gRNA plasmid, and 1 μl ssODN template (10 μM). Then, the resuspended cells were transferred to a nucleocuvette strip chamber and electroporated by using the Amaxa nucleofector, with CM-130 Nucleofector 4D program. After 48 h, GFP⁺ cells were sorted and expanded. Then, single-cell sorting was performed on a FACSAria (BD Biosciences) flow cytometer. Individual clones were splitted, and one part was lysed in 20 μl of the following buffer: 10 mM Tris–HCl, 50 mM NaCl, 6.25 mM MgCl₂, 0.045% NP40, and 0.45% Tween-20 at pH 7.6. After adding 1 μl of 20 μg/ml proteinase K, the lysate was incubated at 56 °C for 1 h and at 95 °C for 15 min. Subsequently, the sample was polymerase chain reaction-amplified using FastStart™ High Fidelity PCR System and specific primers (Supplementary Data S1).

The purity of PCR fragments was checked by agarose gel electrophoresis. Single bands were purified on column using the QIAquick PCR Purification Kit (Qiagen). Purified PCR products were pooled and used for library preparation and deep-sequencing (Galseq). Molecular barcodes were used to deindex the individual amplicons. The presence of the mutations was routinely checked by Sanger Sequencing (GATC Biotech AG).

The TF-1 human erythroleukemia cell line was purchased from DSMZ and maintained in RPMI-1640 medium (Lonza Cambrex) supplemented with 10% FBS (Euroclone), 2 mM L-glutamine (Euroclone), 100 U/ml penicillin (Euroclone), 100 μg/ml streptomycin (Euroclone), 20 mM HEPES (Euroclone), and 5 ng/ml human GM-CSF (Life Technology). Cell were incubated in a humidified atmosphere at 37 °C with 5% $CO_2$, and routinely screened for mycoplasma contamination (GATC Biotech AG). Cell number and viability were assessed by Trypan Blue (Sigma) count every 48 or 72 h.

TF-1-ETNK1-WT, TF-1-ETNK1-243, TF-1-ETNK1-244, and TF-1-Empty were obtained as described in Gambacorti-Passerini et al.[6], and kept under selection in presence of blasticidin 10 mg/ml.

**Deep-sequencing**. Amplicon libraries were generated starting from 500 ng PCR product purified on agarose gel. PCR product was end-repaired and adenylated at 3′ ends before ligation of Truseq DNA Adapter Indices, and then amplified with 6-cycles PCR. Libraries were subsequently sequenced on Illumina HiSeq instrument with paired-end reads 150 bp long. Paired Fastq were initially deindexed using a custom, home-made tool and subsequently aligned to the reference human genome (hg38) using BWA[28]. Bam alignment files were generated from sam using Samtools[29]. Variant calls were performed using CEQer2[30].

**Site-directed mutagenesis and competent cells transformation**. The pCas9-D10A_GFP plasmid (a gift from Kiran Musunuru, Addgene plasmid #44720) was used as a template for in vitro site-directed mutagenesis. Plasmid coding for substitution at position D10A, named pCas9_WT_GFP was generated. Site-

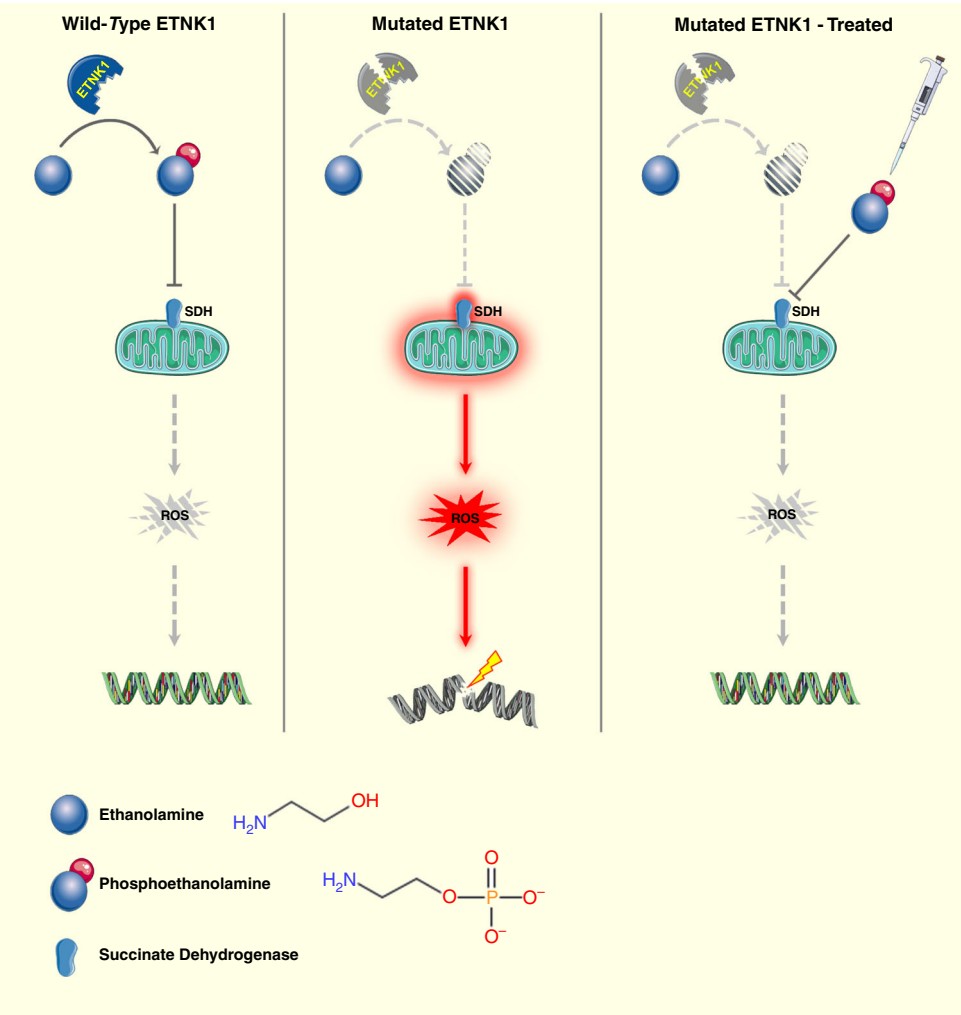

**Fig. 7 Diagram showing the proposed model for ETNK1 mutations.** Left panel: wild-type ETNK1 actively phosphorylates Et leading to the accumulation of P-Et, which in turn negatively modulates mitochondrial activity and ROS production through inhibition of SDH. Middle panel: in the presence of mutated ETNK1 the production of P-Et is impaired, which causes abnormal mitochondrial activation, increased ROS production and DNA damage. Right panel: treatment of ETNK1-mutated cells with exogenous P-Et leads to restoration of normal mitochondrial activity through suppression of SDH, normalization of ROS production and protection of DNA from ROS-mediated damage. Elements of the image were obtained from https://smart.servier.com/ under a Creative Commons Attribution 3.0 License.

directed mutagenesis was conducted using QuikChange II XL Site-Directed Mutagenesis Kit (Stratagene) according to manufacturer's instructions. For site-directed mutagenesis the following oligonucleotide sequences were used:

- Cas9_WT_FW:
  5′–GAAGTACTCCATTGGGCTCGATATCGGCACAAACAGCGTCG–3′
- Cas9_WT_REV:
  5′–CGACGCTGTTTGTGCCGATATCGAGCCCAATGGAGTACTTC–3′

After the digestion with DpnI restriction enzyme (Roche) for at least 1 h at 37 °C, 2 µl of DpnI-treated PCR product were used for transformation of TOP10 competent cells (Life Technology) by heat shock according to manufacturer's instructions. Transformed cells were plated on Luria-Bertani (LB)-ampicillin (50 µg/ml) agar plates and incubated overnight at 37 °C. Bacterial colonies picked from plates were grown overnight at 37 °C in LB with 50 µg/ml ampicillin (Euroclone). Plasmids were recovered from 10 clones using Zyppy plasmid Miniprep Kit (Zymo Research), and the presence of the A10D substitution was confirmed by standard sequencing (GATC Biotech AG), using the following primer:

- Cas9_seq_rev: 5′–GCAGGTAGCAGATCCGATTC–3′

Clones carrying the desired mutation were amplified and plasmid was extracted using NucleoBond Xtra Maxi EF (Macherey-Nagel) and verified again by Sanger sequencing (GATC Biotech AG).

**Quantitative real-time PCR (RT-qPCR).** Five million cells were lysed in TRIzol (Thermo Fisher Scientific) and total RNA was extracted according to manu-facturer's instructions. 1 µg of total RNA was used to synthesize cDNA using

reverse transcription reagents (Thermo Fisher Scientific) after pre-treatment with DNAseI (Thermo Fisher Scientific) to avoid contamination from genomic DNA. Real-Time-Quantitative PCR (RT-qPCR) was performed using Luna® Universal Probe qPCR Master Mix (NEB) on a Stratagene-MX3005P (Agilent Technologies) under standard conditions. The housekeeping glucoronidase beta gene (GUSB) was used as an internal reference. TaqMan® Gene Expression Assays Hs01071698_m1 (Thermo Fisher Scientific) was used.

**Electron microscopy.** Samples were treated for 10 min with 0.12 M sodium phosphate buffer (twice). Then, specimens were fixed for 1 h at room temperature with a mixture of 2% glutaraldehyde, 4% formaldehyde (freshly prepared from paraformaldehyde) in 0.12 M sodium phosphate buffer (pH 7.4). Following the fixation, samples were washed two times in sodium phosphate buffer (0.12 M) for 10 min. Small pellet blocks are post fixed for 1 h at 20 °C with 1% osmium tetroxide in sodium cacodylate buffer (0.12 M). Subsequently, specimens were dehydrated for 10 min with ascending concentration series of ethanol (50%, 70%, 96%, and 100%). After a further incubation with ethanol 100%, cells were treated twice for 10 min with propylene oxide. Specimens were left overnight in 50% propylene oxide and 50% epon 812 under a vented hood. The following day, flat was embedded in pure fresh resin (epon 812). Silver gray sections were cut on an ultracut microtome (Reichert-Jung), double stained with uranyl acetate and lead citrate, and examined in a PHILIPS CM10 electron microscope.

**Single cell force spectroscopy (SCFS).** Prior to experiments, cells were cultured on poly-D-lysine functionalized glass in culture medium, in the CO₂ incubator for

at least 2 h, then washed very carefully two times with PBS solution (pH 7.4). All SCFS measurements were carried out using standard atomic force microscopy (Nanowizard II, JPK) working in force spectroscopy mode. To prevent significant changes in morphology or biochemistry of living cells outside the incubator, each sample was measured within 3 h. During that time, at least two force maps ($8 \times 8$ pixels grid, scan size of $8 \times 8\ \mu m^2$) in the center of the cells were recorded. The force set point was set at 0.5 nN and the approach and retract speeds were kept at 2 $\mu m/s$. For a given cell line type at least 25 living cells were measured. The cantilevers (MLCT-A, Nominal Constant 0.07 N/m, Bruker Corporation) were calibrated before the SCFS measurements both in air and in PBS solution using Thermal Noise method. The force–displacement curves between contact and 200 nm deformation depth have been analyzed, and the softer the sample the larger the deformation recorded. The evaluation of cells elastic properties, described quantitatively through the Young's modulus, was obtained by force curves analysis with the Hertz–Sneddon contact mechanics for a paraboloidal tip[31,32].

**Cell treatment for subsequent immunofluorescence analysis.** Suspension cells (TF-1 and patient's cells): 5 million cells were treated with or without P-Et 1 mM for 24 h. Adherent cells (293 cell line): 150,000 cells were seeded on coverslips previously treated with poly-D-lysine (Sigma) at 0.1 mg/ml concentration. After adhesion, cells were treated with P-Et 1 mM or tigecycline 2.5 µM for 24 h, depending upon experimental conditions.

**Mitochondrial mass and activity.** Mitochondrial mass estimation was performed with MitoTracker Green, while both MitoTracker Red and JC-1 were used to assess mitochondrial activity. MitoTracker assay was performed for 30 min in culture condition with 50 nM MitoTracker Red CMX (Thermo Fisher Scientific), 50 nM MitoTracker Green (Thermo Fisher Scientific), and Hoechst 33342 (Thermo Fisher Scientific). Then, cells were washed in PBS and glass coverslips were mounted on glass slides with a 90% (v/v) glycerol/PBS solution. MitoProbeTM JC-1 assay (Thermo Fisher Scientific) was performed for 30 min in culture condition with 2 µM JC-1, and Hoechst 33342 (Thermo Fisher Scientific). The mitochondrial membrane potential disrupter CCCP (Thermo Fisher Scientific) was used at 50 µM as a control.

**Mitochondrial ROS production.** Cells were incubated for 30 min in culture condition with 5 µM CellROX® Green Reagent (Thermo Fisher Scientific), fixed for 15 min at room temperature in 4% (w/v) paraformaldehyde in 0.12 M sodium phosphate buffer, pH 7.4 then incubated 10 min with Hoechst 33342 (Thermo Fisher Scientific), washed with PBS, and left in PBS overnight at 4 °C. Then glass coverslips were mounted on glass slides with a 90% (v/v) glycerol/PBS solution. ROS production was inhibited with a preincubation with 10 mM N-acetyl-L-cysteine (Sigma Aldrich) resuspended in water for 16 h. Regarding UV damage assay, cells were exposed to UV (cell hood sterilizing lamp) for 30 min, and then kept in culture condition for 2 h before CellROX® incubation.

**γH2AX detection.** Cells seeded on glass coverslips were washed twice with PBS and fixed for 15 min at room temperature in 4% paraformaldehyde in 0.12 M sodium phosphate buffer, pH 7.4, then left in PBS overnight at 4 °C. Cells were incubated for 2 h at room temperature with phospho-Histone H2A.X (Ser139) (Cell Signaling Technology) primary antibody (1:100 dilution in GDB buffer [0.02 M sodium phosphate buffer, pH 7.4, containing 0.45 M NaCl, 0.2% (w/v) bovine gelatin]), followed by staining with Alexa 488-conjugated secondary antibody (1:100 dilution in GDB buffer) (Thermo Fisher Scientific) for 1 h (for further details, please see Supplementary Data 15). After two washes with PBS and staining with Hoechst 33342 (Thermo Fisher Scientific), coverslips were mounted on glass slides with a 90% (v/v) glycerol/PBS solution. For 3D reconstruction of the whole cell volume, a Z-stack of 35 sequential optical planes was acquired and the obtained images were merged using the rendering tool of Zen 2009 software (Carl Zeiss) using manual thresholding.

**Confocal images acquisition.** Images were acquired using Zeiss LSM 710 confocal laser-scanning microscope (Zeiss) using a ×63, 1.4 N/A oil-immersion objective, applying an additional hardware zoom were required. Laser intensities and acquisition parameters were held constant throughout each experiment.

**Mitochondrial respiration.** Mitochondrial respiratory capacity was measured in intact cells (1 million) using high-resolution respirometry (Oxygraph-2k, Oroboros Instruments), in 2 ml glass chambers with stirrer speed 750 rpm. Data were recorded with DatLab 6 software. Correction for instrumental background and air calibration was performed according to the manufacturer's instructions. All respiratory measurements were carried out in the MiR05 buffer (0.5 mM EGTA, 3 mM MgCl₂·6H₂O, 60 mM K-lactobionate, 20 mM taurine, 10 mM KH₂PO₄, 20 mM HEPES, 110 mM sucrose, and 1 g/l BSA (fatty acid free) adjusted to pH 7.1 with 5 M KOH at 37 °C) at normoxia at 37 °C.

The protocol used to measure mitochondrial respiratory capacity was as follows: initially samples were left to stabilize at a routine respiration state. Then two different protocols were performed. O₂ flow was measured after sequentially

addiction of succinate (final concentration of 250 µM), P-Et (titrated at final concentrations of 100 µM, 1, and 2 mM), and malonate (1 mM); or after sequentially addiction of P-Et (titrated at final concentrations of 100 µM, 1, and 2 mM), succinate (250 µM), and malonate (1 mM).

Since it is known that succinate is not able to enter intact cells, the succinate prodrug NV189 was used[33].

**Cells preparation and mitochondria isolation for lipidomics.** Cells were collected and washed twice in PBS without Mg++ and Ca++, to reach concentration of 1500 cells/µl, in 1 ml, stored at −80 °C, and shipped to Lipotype GmbH.

Mitochondria were isolated using Mitochondria Isolation Kit for Cultured Cells (Thermo Fisher Scientific), Option B with dounce homogenization, according to manufacturer's instructions. Briefly, 20 million of cells harvested, resuspended in 800 µl Reagent A supplied, and transferred to pre-chilled dounce tissue grinder. After 80 strokes Reagent C was added, and sample was centrifuged at $700 \times g$ for 10 min at 4 °C. The surpernatant was further centrifuged at $3000 \times g$ for 15 min at 4 °C and the mitochondria pellet was resuspended in PBS to reach the final concentration of 0.35 mg/ml, stored at −80 °C, and shipped to Lipotype GmbH.

**Lipid extraction for mass spectrometry lipidomics.** Lipid analysis was performed by Lipotype GmbH using mass spectrometry as previously described[34]. Lipids were initially extracted using a chloroform/methanol extraction protocol[35]. Subsequently, each sample was spiked with a mixture of lipids containing: cardiolipin 16:1/15:0/15:0/15:0 (CL), ceramide 18:1;2/17:0 (Cer), diacylglycerol 17:0/17:0 (DAG), hexosylceramide 18:1;2/12:0 (HexCer), lyso-phosphatidate 17:0 (LPA), lyso-phosphatidylcholine 12:0 (LPC), lyso-phosphatidylethanolamine 17:1 (LPE), lyso-phosphatidylglycerol 17:1 (LPG), lyso-phosphatidylinositol 17:1 (LPI), lyso-phosphatidylserine 17:1 (LPS), phosphatidate 17:0/17:0 (PA), PC 17:0/17:0, PE 17:0/17:0, phosphatidylglycerol 17:0/17:0 (PG), phosphatidylinositol 16:0/16:0 (PI), phosphatidylserine 17:0/17:0 (PS), cholesterol ester 20:0 (CE), sphingomyelin 18:1;2/12:0;0 (SM), triacylglycerol 17:0/17:0/17:0 (TAG), and cholesterol D6 (Chol). Following extraction, the sample (organic phase) was dried using a speed vacuum concentrator. The first-step dry extract was re-suspended in 7.5 mM ammonium acetate in chloroform/methanol/propanol (1:2:4, V:V:V) and the second in 33% ethanol solution of methylamine in chloroform/methanol (0.003:5:1; V:V:V). Liquid handling was performed with Hamilton Robotics STARlet platform in combination with the Anti-Droplet Control feature.

**Mass spectrometry data acquisition.** Data acquisition was performed using a QExactive mass spectrometer (Thermo Fisher Scientific) in combination with a TriVersa NanoMate ion source (Advion Biosciences). Positive and negative ion modes were used to analyze the biological samples, using a resolution of R$m/z$ = 200 = 280,000 for MS and R$m/z$ = 200 = 17,500 for MS/MS experiments, in a single acquisition. An inclusion list encompassing corresponding MS mass ranges scanned in 1 Da increments was used to trigger MS/MS[36] to monitor CE, DAG, and TAG ions as ammonium adducts; PC, PC O-, as acetate adducts; and CL, PA, PE, PE O-, PG, PI, and PS as deprotonated anions both MS and MS/MS data were used. LPA, LPE, LPE O-, LPI, and LPS as deprotonated anions, Cer, HexCer, SM, LPC, and LPC O- as acetate adduct and cholesterol as ammonium adduct of an acetylated derivative were monitored using MS[37].

**Data analysis and post-processing.** All the MS data were analyzed by using custom software based on LipidXplorer[38,39]. A dedicated data management system was used to perform normalization and post-processing of raw data. As filters for lipid identification a signal-to-noise ratio > 5, and a signal intensity 5 fold higher than in blank samples were applied.

**Targeted mass spectrometric metabolite profiling and quantification of intracellular P-Et and serine.** ETNK1-WT, ETNK1-N244S, and ETNK1-KO cells were seeded (6 million cells/dish) and incubated for 48 h to reach ~70% confluency. Three to four replicate cultures were sampled as described by Kvitvang et al.[40]. Following mechanical detachment, cells were immediately quenched in LN₂. Sampled cells were extracted by three repeated freeze–thaw cycles in water/acetonitrile (1/1, v/v). Extracts were up-concentrated by lyophilization and reconstituted in solvents compatible with downstream MS analysis.

Phosphorylated metabolites and organic acids were quantified by capillary ion chromatography tandem mass spectrometry (capIC-MS/MS) as described in refs. [40,41], and by LC–MS/MS with up-front derivatization as described in ref. [42], respectively.

P-Et and serine were quantified by LC–MS/MS. The analysis was performed on an ACQUITY I-Class UPLC (Waters) coupled to a Xevo TQ-XS triple quadrupole mass spectrometer (Waters) equipped with an electrospray source operating in positive mode. Samples (2 µl) were injected onto a SeQuant ZIC-cHILIC 100 × 2.1 mm column with a pore size of 3 µm (150657, Merck). The column was maintained at 10 °C and eluted with mobile phases (A) H₂O and (B) H₂O/ACN 1/4 (v/v), both added 20% 100 mM ammonium acetate (v/v), pH 8. The following gradient was applied at a flow rate of 200 µl/min: 0–1 min: 70% B, 1–5.5 min: 70–40% B, 5.5–6 min: 40–70% B, 6–10 min: 70% B. P-Et was quantified from the precursor–product

ion transition: $m/z$: 142–44, CV: 16 V, CE: 8 eV. Serine was quantified from the precursor–product ion transition: $m/z$: 106–60, CV: 18 V, CE: 8 eV.

MS data processing was performed in TargetLynx application manager of MassLynx 4.1 (Waters). Absolute quantification was performed by interpolation from calibration curves prepared from serial dilutions of respective analytical grade standards calculated by least-squares regression with $1/x$ weighting. For metabolite profiling, extract concentrations were corrected for dilutions and normalized to cell density at the time of sampling to yield mole/cell. Extract P-Et levels were normalized to extract serine levels.

**Quantification of extracellular glucose and lactate**. Extracellular glucose and lactate in fresh and spent medium from ETNK1-WT and ETNK1-N244S cultures was quantified by recording 1D proton NMR spectra and applying the electronic reference to access in vivo concentrations (software: ERETIC2, Topspin 3.5, Bruker) as described in ref. [43]. Consumption/production was normalized to the average number of live cells in the 48 h interval to obtain consumption/production/cell/48 h.

**ChIP-seq analysis of 8-oxoguanine distribution**. Five million cells were harvested and DNA was extracted using PureLink Genomic DNA kit according to the standard protocol (Thermo Fisher Scientific). 7 μg of DNA in a final volume of 150 μl were sonicated with a Bioruptor sonicator system (Diagenode), gel purified with QIAquick PCR Purification Kit (Qiagen), and subsequently immunoprecipitated with Anti-8-Oxoguanine Antibody, clone 483.15 (Millipore-Merck) 5 μl/reaction and Pierce Protein A/G Magnetic Beads (Thermo Fisher Scientific). After immunoprecipitation, DNA was purified and libraries were prepared for sequencing following the Illumina ChIP-Seq protocol (TrueSeq ChIP library prep kit IP-202-1012) with an Illumina HiSeq4000 in paired-end mode (Galseq).

**Oxoguanine analysis**. Paired Fastq files were initially mapped to the reference human genome (hg38) using BWA[28]. Bam were generated from sam alignment files using Samtools[29]. Subsequently, sorted, indexed Bam files were processed by a custom home-made tool (OxoGuanineProfiler). Briefly, individual Bam files were filtered for reads with mapping quality ≥ 20. Reads mapping to mitochondria chromosome were similarly discarded. The remaining reads were binned according to their first-base position in the reference genome, using 10,000-bases bin size. All the bins were subsequently normalized for the total number of filtered reads. Bins containing <10 hits were discarded. Finally, the total number of normalized hits per individual chromosome was calculated for two independent experiments and the mean value calculated. The final statistical analysis was performed by OxoGuanineProfiler on mean values using a Wilcoxon signed-rank test, with threshold $p$-value set at 0.05.

**6-TG assay**. Cells were cultured in presence of P-Et 1 mM and tigecycline 2.5 μM for 15 days, refreshed every 2 days. Then 1 million of cells was plated and exposed to 30 μM of 6-TG for 15 days. In parallel, as control, 1500 cells were plated for 15 days. Colonies were counted and the induced mutant frequency was calculated as following:

$$\frac{\text{total number of 6} - \text{TG resistant colonies}}{1,000,000} \bigg/ \frac{\text{total number of untreated colonies}}{1500}. \tag{1}$$

Images were acquired using Zeiss Primovert light microscopy. Colonies area was counted using specific homemade-designed macro with ImageJ.

**Patients**. Diagnosis of aCML and related diseases was performed according to the WHO 2008 and 2016 classifications. All patients provided written informed consent, which was approved by the ethics committee of the University of Milano - Bicocca. This study was conducted in accordance with the Declaration of Helsinki. Sample collection and processing, and exome sequencing were performed as already described[44].

**Interphase-FISH**. Interphase FISH was performed on genome-edited and WT 293 lines as previously described[45]. Genomic probe for *ETNK1* gene (RP11-268P4/12p12.1—UCSC Genome Browser February 2009, GRCh37/hg19 Assembly: 22,707,126–22,881,094), was labeled with spectrum orange (Vysis) and combined with green alpha satellite centromeric probe of chromosome 12 (Vysis, Abbott Laboratories; Cytocell Ltd).

The ploidy was estimated on the mean signal numbers of centromeric probes. Nuclei with a target/centromeric probe ratio of ≥3:1 were defined as amplified, and those with a probe ratio of >1:1 but <3:1 were classified as relative copy gain. Nuclei with a probe ratio 1:1 but more than two copies of each probe were defined as polisomy. Analyses were carried out on 200 nuclei/experiment.

**RNA-sequencing**. Five million cells were lysed in TRIzol (Thermo Fisher Scientific) and total RNA was extracted according to manufacturer's instructions. 4 μg of RNA (concentration 400 ng/μl) were used for library preparation (Galseq).

**Western blot and antibodies**. Five million cells were seeded in absence or presence of P-Et for 24 h. Then cells were harvested, washed once in PBS at 4 °C, and resuspended in 100 μl Laemmli buffer supplemented with 10% β-mercaptoethanol (Sigma). Lysates were denatured at 99 °C for 20 min and then used for electrophoresis. Equal volumes (30 μl) were loaded on 12% sodium dodecyl sulfate polyacrylamide gel electrophoresis (SDS–PAGE), transferred to nitrocellulose membrane Hybond ECL (GE Healthcare Life Science), and incubated overnight at 4 °C with primary antibody (1:1000 dilution in BSA 2.5%, Roche). Secondary horseradish peroxidase-conjugated anti-rabbit and anti-mouse antibodies (1:2000) were incubated for 1 h at room temperature and then bands were visualized by chemiluminescence ECL (Thermo Scientific) as recommended by the manufacturer. Phospho-Histone H2AX (Ser139), ETNK1, β-Catenin, H3, actin, anti-mouse, and anti-rabbit antibodies were purchased from Cell Signaling Technology, GeneTex, BD, Abcam, Sigma, and Bio-Rad, respectively. For further details regarding antibodies, please see Supplementary Data 15. For uncropped and unprocessed scans please see Source Data file.

**Mitochondrial complexes I, II, and IV activity**. 180 million of cells were harvested, and resuspended first in NKM buffer (1 mM Tris–HCl, pH 7.4, 0.13 M NaCl, 5 mM KCl, 7.5 mM MgCl$_2$), and later in homogenization buffer (10 mM Tris–HCl, 10 mM KCl, 0.15 mM MgCl$_2$, 1 mM AEBSF, 1 mM DTT). Cells were transferred to a pre-chilled glass homogenizer, incubated for 10 min on ice, and homogenized. The homogenate was transferred into a conical centrifuge tube containing 2 M sucrose solution, and centrifuged at $1200 \times g$ for 5 min. This treatment was repeated twice. Mitochondria were collected by centrifugation at $7000 \times g$ for 10 min and resuspended in mitochondrial suspension buffer (10 mM Tris–HCl, pH 6.7, 0.15 mM MgCl$_2$, 0.25 mM sucrose, 1 mM AEBSF). Mitochondria were lysed on ice for 30 min with the supplied 10× detergent solution, and centrifuged at 4 °C at $12,000 \times g$. Surnatant was collected and used for the measurement of mitochondrial activity according to manufacturer's instructions. The following kits (Abcam) were used: Complex I Enzyme Activity Microplate Assay Kit (Colorimetric) (ab109721), Complex II Enzyme Activity Microplate Assay Kit (ab109908), and Complex IV Human Enzyme Activity Microplate Assay Kit (ab109909).

**Mitochondrial complex III activity**. Mitochondria were extracted as described above and resuspended mitochondrial suspension buffer without being lysed. Complex III was measured by using Mitochondrial Complex III Activity Assay Kit (BioVision), according to manufacturer's instructions.

**Software and data analysis**. Validation of the CRISPR/Cas9 mutations was performed by ultradeep-sequencing, as already described. Chromatograms were visualized using Chromas 2 (Technelysium). Statistical analysis and graphs were analyzed using GraphPad Prism6 (GraphPad Software, Inc.). Confocal microscopy fields were analyzed using specific homemade-designed macro with ImageJ (https://imagej.nih.gov/ij/) software. Mitochondrial activity quantification was performed measuring the integrated density (ID) of red over green signal ratio, after proper manual thresholding. ROS production was analyzed measuring the ID in nuclear compartment; for γH2AX assay number of foci was detected after manual thresholding and with the precast analyzed particle plug-in of ImageJ. All the data obtained derived from at least 10 fields per experimental condition (at least 200 cells each). Western blot bands were visualized using ChemiDoc™ XRS+ (Bio-Rad).

**Docking**. The structure of P-Et was optimized by the PM3 Hamiltonian in the MOPAC16 package v17.349L (http://openmopac.net (2016)) and docked into the catalytic site of *E. coli* SDH (PDB: 1NEN[46]) by the glide software[47]. For P-Et, the PM3 charges were used for docking. Prior to docking, the SDH structure was processed by the Maestro software (https://www.schrodinger.com/maestro) to add all hydrogens according to the predicted p$K_a$ of amino acids. Before fixing the docking setup, a number of conditions were probed. Shown results refer to docking in the presence of the FAD co-factor upon removal of all water molecules. According to the Glide setup, the protein is rigid whereas the ligand is fully flexible. The five best docking poses were saved, showing the first one.

**Availability of biological materials**. Cell lines used in this study are all available upon request. Availability of primary leukemic samples is however limited, due to the rarity and limited amount of these biological materials.

**Reporting summary**. Further information on research design is available in the Nature Research Reporting Summary linked to this article.

## Data availability

The NGS data discussed in this publication have been deposited in NCBI's Sequence Read Archive (SRA) and are accessible through accession number PRJNA501862. ChIP-Seq bed files, custom software and Oroboros raw data are available at https://osf.io/fd3m2. Source data are provided with this paper.

## Code availability

The scripts used for Oxoguanine Analysis are available from the authors upon request.

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

## Acknowledgements

We kindly acknowledge the contributions of Michela Viltadi, Cristina Mastini, and Silje Malene Olsen for technical help. We thank Sarah Piel and Eskil Elmer for providing succinate prodrug NV189. This work was supported by Fondazione AIRC per la Ricerca sul Cancro 2018 (IG-22082) to R.P., Fondazione AIRC per la Ricerca sul Cancro 2015 (IG-17727) to R.P., Fondazione AIRC per la Ricerca sul Cancro 2017 (IG-20112) to C.G.-P.

## Author contributions

D.F.: Investigation, methodology, visualization, writing—review & editing, M.M.: Investigation, methodology, visualization, R.R.: Investigation, M.D.: Investigation, I.C.: Investigation, L.M.R.: Investigation, M.J.: Investigation, A.N.: Investigation, D.D.: Investigation, L.M.: Formal analysis, M.B.e.: Investigation, G.Z.: Investigation, M.B.o.: Investigation, S.C.: Resources, B.C.: Resources, F.F.: Investigation, formal analysis, V.C.: Investigation, formal analysis, R.C.: Investigation, formal analysis, D.S.: Investigation, formal analysis, L.N.: Investigation, formal analysis, C.C.: Investigation, formal analysis, F.Man.: Supervision, C.M.: Supervision, F.Mag.: Supervision, G.C.: Supervision, P.B.: Formal analysis, supervision, D.R.: Resources, S.L.: Investigation, formal analysis, Supervision, C.G.P.: Conceptualization, funding acquisition, writing—review & editing, supervision, R.P.: Conceptualization, methodology, funding acquisition, writing—original draft preparation, supervision.

## Competing interests

The authors declare no competing interests.
