## [Peer Review File · Nature Communications]

Reviewers' comments:

Reviewer #1 (Remarks to the Author):

Fontana et al., have evaluated the functional effects of ETNK1 mutations. ETNK1 regulates the production of the phospholipid P-Et. Knockout and mutations of ETNK1 lead to increased mitochondrial mass, increased oxygen consumption and ROS production. ETNK1 inhibition should lead to decreased P-Et and supplementing respiratory complexes with P-Et impaired respiratory chain complex II activity.

The authors report the novel finding that P-Et impairs complex II activity likely be competing with succinate for binding to SDH. This work sheds new light on the regulation of complex II.

I have the following comments about the manuscript:

1) The vast majority of experiments were conducted in 293 cells and only a few experiments were validated in a leukemia cell line. It is unclear how the findings from these cell lines would translate to patient samples. Specifically, it is unclear whether primary samples with ETNK1 mutations or deletions have metabolic abnormalities and alterations in P-Et levels. It is also unknown whether they have abnormalities in mitochondrial and respiratory chain complex function. As such, the relevance of their findings to human disease is unclear.

2) The authors did not measure levels of P-Et in their cell lines, Does knockout or mutation of ETNK1 alter P-Et levels? Of note, mutations and knockout of ETNK1 did not alter other lipid levels.

3) What are the lipid levels in patient samples with this mutation?

4) A potential implication of their work is that patients with ETNK1 should be treated with P-Et to restore normal mitochondrial function. However, without mouse models and primary samples, this question has not been tested.

5) The authors treat cells with mM concentrations of P-Et to see cellular effects. Are these concentrations physiologically relevant?

6) The authors treat complexes with uM concentrations to see changes in activity. But, changes in cells require mM concentrations. Can the authors be certain that the effects on mitochondrial function are only due to alterations in respiratory chain complex II

7) The authors should show immunoblotting to demonstrate target knockout with CRISPR.

8) As a minor comment, the manuscript results and figure legends should specify which cells were used. One had to refer to the methods to determine the experiments were performed in 293 cells. Likewise, times of incubation should be noted in the text.

Reviewer #2 (Remarks to the Author):

These studies investigate the mechanisms of cellular dysregulation caused by catalytically inactivating mutations in ETNK1 in leukemias. Although these mutations are recurrent in aCML and CMML, very little is known about the mechanisms by which they contribute to oncogenesis, making these studies of biological interest. Experiments showing altered mitochondrial activity, ROS, and H2AX levels with ETNK mutations and restoration of appropriate levels of activity upon P-Et supplementation are compelling and mechanistically interesting. The finding of increased H2AX levels in ETNK1 mutant aCML samples adds nice clinical relevance to this study. It is very interesting the ETNK1 mutation are early events in disease evolution, however it is hard to evaluate the validity of these claims based on the data presented. There are some improvements that need to be made with respect to statistical analysis and data interpretation, but otherwise it is an interesting and novel story.

Major comments-

1. Lines 132-135: I would be reluctant to point out changes in gene expression that are not significant. Furthermore, I'm not sure what biological relevance such a small difference in expression of a biosynthetic enzyme would have. In general, it takes fairly large changes to

enzyme abundance to alter function. This portion of the manuscript could be omitted without significantly altering the conclusions.

2. Statistical analyses in many figures with multiple groups use a t-test. This may not be an appropriate test in the setting of multiple comparisons. The authors should consult a statistician, but an Anova with a post-test may be a more appropriate analysis.

3. For figure 4B the changes in oxoG seem very subtle with the exception of chr18...honestly, I'm a bit surprised that the differences are statistically significant and seem subtle relative to the change in level of dsDNA damage. Can the authors provide some additional information about the global magnitude of this change? Are there potential oxoguanine-independent mechanisms by which dsDNA damage could be occurring downstream of mutant ETNK1?

4. Line 316-322. I see the data for the bulk based sequencing to assign the mutations with oncogenic potential in supplemental tables 12-16, but don't see a reference to where the data from the sequencing of individual clones is presented used to assign the hierarchies depicted in 6A-D. It seems like this data should be in the supplement, but I am unable to find it. How does the reconstruction by mutational analysis of individual colonies compare to the assignment that would have been made based on VAF?

Minor-

5. Line 53- synthesize (spelling)

6. Lines 71-75- run on sentence

7. Figures 1C-E. While it is easy to appreciate that there are areas of low electron density in the mutants it is not easy to appreciate that the WT mitochondria have a rounder shape. Maybe just omit that statement, or alternatively present a measurement of circularity.

I believe that the level of magnification is consistent between the images, but the scale bar is a bit challenging to see (especially in E), perhaps make the scale bar more prominent.

8. Line 124 and 138. I don't understand what is meant by vicariate in this particular context.

9. Figure S3. Can the font size be increased on the legends? What do the green bars represent?

10. Figure S4. A brief description of the method used in the figure legend would be helpful.

11. Line 203. It would be more accurate to say that it restored to near normal levels

12. Line 218. I would soften the statement that it "almost completely restored the normal ROS levels" perhaps to "reduced ROS levels to an intermediary level" or something similar. The levels of ROS are still fairly elevated relative to baseline.

13. Line 243. A sentence or two explaining the 6-TG assay would be helpful here.

14. Line 250. I may just be misreading this section, but if the 6-TG resistance assays were performed with P-ET and tigecycline as new assays from those described in line 245 they should be shown separately and not as a combined graph ie untreated controls should be from the same experiment. It looks like the control data is identical between supplemental figure 4 and 5...so I can't tell why they are presented as two different tables, unless the untreated is being shown for reference, but was not part of the second experiment.

15. Line 266. What is the evidence for aCML being characterized by chromosomal instability? A

reference would be useful here.

16. 366-374 The discussion of the time involved to do functional studies on newly identified mutations does not seem necessary to me.

17. 396-397. I believe that it would be more correct to say "dominant" than "dominant negative". Dominant negative implies that the mutant protein interferes with the activity of WT ETNK1, which I don't think has been demonstrated here.

Reviewer #3 (Remarks to the Author):

Summary of the work:

The authors have evaluated the effect of mutations in the kinase ETNK1 on mitochondrial activity in this paper. Using CRISPR KO and knock-in 293 HEK cell lines, they show that loss of ETNK1 has no significant effect on phospholipid levels in the mitochondria or in the cell membrane, suggestive of mechanisms compensating for loss of ETNK1 enzymatic activity. However, they note that ETNK1 loss leads to increased mitochondrial mass, ROS production and Oxo-DG and gH2AX, which is hypothesized to be due to a regulatory mechanism of mitochondrial complex II by phosphoethanolamine. Importantly, they demonstrate that the increased ROS and Oxo-DG in ETNK1 mutant cells is reversible by p-ET addition.

Overall I found the discoveries to be interesting, and relevant/ topical. However, the findings need to be defined further in a suitable cellular model, the link between ETNK1 loss and DNA damage needs to be clarified, and the presentation of data needs to be significantly improved in its rigour/ attention to methodological detail.

Major comments:

1. The majority of the experiments are based on ETNK1 knockout/knockin in 293 HEK cells. Given the significant differences in phenotype of mutations in distinct cellular contexts, I think it is required to carefully validate all critical findings in a leukemic or haematopoietic cell line, to justify the conclusion of ETNK1 regulating leukemogenesis through metabolism. The overexpression experiments in TF1 cells suggest that mutant ETNK1 can act as a dominant negative, but there is no data presented on the extent of overexpression that leads to this effect (the mutations in aCML are noted to be heterozygous).
2. The cell line models are not adequately validated. Do the authors see a change in p-ET in their 293 KO/ knockin cells, similar to their previous observations? These data should be presented along with actual FISH analysis, sequencing, and western blots to show validation of the knockout and knockin in the 293 cell line model (along with validation of the overexpression system in the haematopoietic line).
3. For all analysis represented in the paper, the authors are requested to specify the imaging/ quantification technique used, and the number of cells analyzed per experiment. Eg. Mitochondrial mass estimation has been performed with mitotracker green, but the methods and figure legend do not elaborate how the analysis was performed. The authors should also provide a table with all the antibodies used (clone/ product numbers), and concentration of primary antibody for each application.
4. ETNK1 mutations are hypothesized to reduce p-ET levels, leading to reduced "brakes" on SDH activity, and therefore increased ROS production from complex II, and subsequent DNA damage. Is there evidence of increased complex II activity in lysates from the ETNK1 KO/ knockin cells? Do these cells show evidence of altered respiration in a seahorse assay?
5. If ROS generation is key to ETNK1 mutations in leading to genomic instability, does reduction of ROS reverse it? Would quenching of ROS (eg. by N-acetyl cysteine) reduce the amount of Oxo-DG and/or gH2AX? This would be pertinent given recently described links between SDH mutations and impaired homologous recombination, as alternate means of leading to elevated gH2AX/ genomic

instability.

6. The section on ETNK1 being an early/ truncal event in leukemogenesis does not fit into the current flow of the paper. It could serve as an introductory section if required, or could be removed from this paper.

Specific comments:

1. The Figure 1 title reads "Mitochondria morphology and activity". The authors use mitotracker green/red ratio for mitochondrial potential. As this is a key message of the paper, it would be good to cross-validate these findings with an orthogonal assay such as TMRE or JC1 fluorescence, in both the CRISPR KO and overexpression system. Positive and negative controls are important to include for these experiments, for example FCCP/CCCP treatment to demonstrate depolarization (Figure 1 and 2); ROS quenchers and inducers (Figure 3B).
2. In Figure1 (panel C, D, E) authors show TEM images of mitochondria suggesting an abnormal mitochondrial morphology in ETNK1 mutant and KO cell lines. Are these accompanied by/ due to changes in cellular morphology?
3. In figure 2 (and throughout the paper), please report actual p-values when t-tests are done. How many cells were analyzed for each experiment and using what method?
4. How does the in-vitro dose of Tigecycline of 2.5 microM compare to the pharmacokinetics of Tigecycline dosage in humans?
5. In figure 3, why were only WT and KO cells (and not the mutant) chosen for the O₂ consumption experiments? How many cells were analyzed per experiment for the CellRox experiments? How were the imaging data quantified?
6. Does ETNK1 loss/ mutation lead to increased overall OxoDG staining (as assessed by immunofluorescence)? For the 6TG assay (Figure 4D); please show representative images in the main or supplementary data
7. gH2AX is smaller than Actin, and ideally should be presented below it in a western blot figure. The quality of the blots is not satisfactory (especially for in-vitro samples).
8. Figure 5B is not representative at present. Please highlight an area from figure 5A and enlarge for clearer view.
9. How were three replicates performed for figure S10 and 5I (single patient sample)? Were they performed at three different occasions or processed simultaneously? How was the p-value calculated here?
10. "side pathways" could be phrased better

Anand D Jeyasekharan
National University of Singapore

We are glad that the three Reviewers found merits in our work and we did our best to address their points. The point-by-point answers to the Reviewers are reported here below.

Answers to Reviewer #1

Fontana et al., have evaluated the functional effects of ETNK1 mutations. ETNK1 regulates the production of the phospholipid P-Et. Knockout and mutations of ETNK1 lead to increased mitochondrial mass, increased oxygen consumption and ROS production. ETNK1 inhibition should lead to decreased P-Et and supplementing respiratory complexes with P-Et impaired respiratory chain complex II activity.

The authors report the novel finding that P-Et impairs complex II activity likely be competing with succinate for binding to SDH. This work sheds new light on the regulation of complex II.

I have the following comments about the manuscript:

1) The vast majority of experiments were conducted in 293 cells and only a few experiments were validated in a leukemia cell line. It is unclear how the findings from these cell lines would translate to patient samples. Specifically, it is unclear whether primary samples with ETNK1 mutations or deletions have metabolic abnormalities and alterations in P-Et levels. It is also unknown whether they have abnormalities in mitochondrial and respiratory chain complex function. As such, the relevance of their findings to human disease is unclear.

To address the appropriate Reviewer's concerns, we extended the entire analysis to a second cell line (TF-1), which is hematological and myeloid and, as such, much closer to the cell type involved in the disease than 293 (Fig 2, Suppl Fig S11, S12, S13, S19, S22 and corresponding text). Also, whenever possible taking into account the rarity of the primary samples, we extended the study to primary ETNK1^{+/-} aCML samples (Fig 5, Suppl Fig S8, S20). Specifically, we replicated on the myeloid TF-1 line the following experiments:

- *mitochondria activity with MitoTracker Red and Green in absence/presence of P-Et;*
- *mitochondria activity with JC1 plus/minus CCCP;*
- *ROS production in absence/presence of P-Et;*
- *γ H2AX Double-strand DNA damage in the absence/presence of P-Et;*
- *Mitochondria Complex I, II, III and IV activity in the absence/presence of increasing concentration of P-Et;*
- *succinate competition assay for Mitochondria Complex II.*

On primary samples, we performed the following new analyses:

- *membrane Lipid class composition analysis;*
- *ROS production in absence/presence of P-E;t*
- *γ H2AX Double-strand DNA damage in absence/presence of P-Et.*

All the new experiments went in the same direction as the previous ones, therefore, strengthening the conclusions of the work.

2) The authors did not measure levels of P-Et in their cell lines, Does knockout or mutation of ETNK1 alter P-Et levels? Of note, mutations and knockout of ETNK1 did not alter other lipid levels.

We thank the Reviewer for pointing out this very important question. To address this point, we extended our previous analysis in order to measure: 1) intracellular P-Et levels (Suppl Fig S10); 2) total lipid levels (Suppl Fig S5); and 3) detailed lipid composition, with specific but not exclusive emphasis on phosphatidylethanolamines (Suppl Fig S6).

Following this new analyses, we can state that: 1) P-Et levels are decreased in mutated and KO ETNK1, as already shown in patients (*Blood*. 2015 Jan 15;125(3):499-503). 2) Total lipid levels don't change across the different genotypes. 3) P-Et levels don't change across the different genotypes, therefore, supporting the idea that intracellular P-Et concentration doesn't represent the limiting factor for P-Et synthesis.

3) What are the lipid levels in patient samples with this mutation?

In line with the points discussed in (2), we extended the same analysis to patient samples. We found that, in line with the cell models, lipid levels don't change in ETNK1+ vs ETNK1- patients (Suppl. Fig. S8).

4) A potential implication of their work is that patients with ETNK1 should be treated with P-Et to restore normal mitochondrial function. However, without mouse models and primary samples, this question has not been tested.

We totally agree with the Reviewer. This work, however, is intended as a first study focused aimed at inferring the molecular mechanisms responsible for the ETNK1-mediated prooncogenic process occurring in aCML patients. The potential of P-Et as a therapeutic agent, which emerged here, will be the subject of future investigations using *in vivo* models. Remarkably, these models are quite complex, given that the role we propose for mutated ETNK1 is to induce a very peculiar 'mutator phenotype' in the target cells and the definition of a valid readout for such a model is not trivial. The translational potentialities of the study will be addressed in the near future.

5) The authors treat cells with mM concentrations of P-Et to see cellular effects. Are these concentrations physiologically relevant?

It is very difficult to say, as the data available in the literature are very limited. Recent data generated in mouse (*Cell Rep*. 2019 Oct 1;29(1):89-103.e7. doi: 10.1016/j.celrep.2019.08.087) suggest that those concentrations may be physiologically relevant.

6) The authors treat complexes with uM concentrations to see changes in activity. But, changes in cells require mM concentrations. Can the authors be certain that the effects on mitochondrial function are only due to alterations in respiratory chain complex II

This is a very good point. A statistically significant effect exerted by P-Et on mitochondrial complexes was detectable only for complex II, indicative of a specific effect of P-Et on this complex. Also, the central role of succinate in the activity of mitochondrial complex II is well known and our finding that increased succinate concentrations compete with P-Et, by restoring complex II activity, further support this notion..As a further support to this evidence, the same experiments, now replicated on myeloid TF-1 cells, produced consistent results with the previous ones.

7) The authors should show immunoblotting to demonstrate target knockout with CRISPR.

We tried to generate publication quality western blots of our CRISPR lines with several commercial anti-ETNK1 antibodies (e.g. Ethanolamine kinase (U-23): sc-130754, SANTA CRUZ BIOTECH, INC. and ETNK1 Polyclonal Antibody: A58978, Epigentek). Unfortunately the quality of these antibodies is so poor that we couldn't be able to provide the requested data. To overcome, at least in part, this limitation, we performed Q-PCR experiments showing that ETNK1 is expressed at similar transcript levels in the WT and N244S lines (Suppl. Fig. S2; $p > 0.05$), while it is four-fold less expressed in the knock-out line compared to the WT one (Suppl. Fig. S2; $p = 0.0033$). This finding is compatible with an expected nonsense-mediated RNA decay occurring in the knock-out.

8) As a minor comment, the manuscript results and figure legends should specify which cells were used. One had to refer to the methods to determine the experiments were performed in 293 cells.

Likewise, times of incubation should be noted in the text.

In order to improve the manuscript readability, we modified the text and legends by specifying both the cell line used and the time of incubation of all the experiments.

Answers to Reviewer #2 (Remarks to the Author):

These studies investigate the mechanisms of cellular dysregulation caused by catalytically inactivating mutations in ETNK1 in leukemias. Although these mutations are recurrent in aCML and CMML, very little is known about the mechanisms by which they contribute to oncogenesis, making these studies of biological interest. Experiments showing altered mitochondrial activity, ROS, and H2AX levels with ETNK mutations and restoration of appropriate levels of activity upon P-Et supplementation are compelling and mechanistically interesting. The finding of increased H2AX levels in ETNK1 mutant aCML samples adds nice clinical relevance to this study. It is very interesting the ETNK1 mutation are early events in disease evolution, however it is hard to evaluate the validity of these claims based on the data presented. There are some improvements that need to be made with respect to statistical analysis and data interpretation, but otherwise it is an interesting and novel story.

Major comments-

1. Lines 132-135: I would be reluctant to point out changes in gene expression that are not significant. Furthermore, I'm not sure what biological relevance such a small difference in expression of a biosynthetic enzyme would have. In general, it takes fairly large changes to enzyme abundance to alter function. This portion of the manuscript could be omitted without significantly altering the conclusions.

We thank the Reviewer for providing this valuable suggestion. We decided to completely omit this section of our work, accordingly.

2. Statistical analyses in many figures with multiple groups use a t-test. This may not be an appropriate test in the setting of multiple comparisons. The authors should consult a statistician, but an Anova with a post-test may be a more appropriate analysis.

We agree that multiple group tests should be addressed using Anova rather than individual t-test. We modified the analyses, figures and text accordingly.

3. For figure 4B the changes in oxoG seem very subtle with the exception of chr18...honestly, I'm a bit surprised that the differences are statistically significant and seem subtle relative to the change in level of dsDNA damage. Can the authors provide some additional information about the global magnitude of this change? Are there potential oxoguanine-independent mechanisms by which dsDNA damage could be occurring downstream of mutant ETNK1?

The limited difference is, in the opinion of the authors, in large part explained by the extremely volatile nature of the oxoG signal. When oxoG is generated at gDNA level, the error must be corrected very quickly, before the onset of the next cell cycle, in order to avoid the irreversible incorporation of the error in one of the daughter cells. Also, when the oxoG derivative is erased by the OGG1 repair machinery, the signal of the existing DNA damage is lost forever. Conversely, in dsDNA damage, the presence of γ H2AX phosphorylation survives for a very long time even after the DSB has been rejoined, as it slowly decays over time (Nucleic Acids Res 36: 5678–5694. 10.1093/nar/gkn550). This is a consequence of the γ H2AX phosphorylation taking place in a very large chromatin region surrounding the DSB and amplifying the DSB signal itself.

The statistical analysis was done using a paired, non-parametric test and should be easily reproduced using the following ChIP-Seq data:

Chromosomes	Mean_N244S	Mean_Par
Chr1	1370.2	1238.3
Chr2	1152.1	1038.1
Chr3	780.5	779.9
Chr4	285.0	209.5
Chr5	228.3	203.3
Chr6	364.6	380.6
Chr7	631.4	557.4
Chr8	547.9	433.5
Chr9	399.4	338.3
Chr10	120.5	132.7
Chr11	256.7	217.0
Chr12	386.1	325.9
Chr13	280.2	181.5
Chr14	214.4	256.5
Chr15	608.3	631.1
Chr16	429.5	450.3
Chr17	127.1	118.2
Chr18	1492.0	1054.3
Chr19	166.8	161.5
Chr20	21.8	33.8
Chr21	32.4	36.1
Chr22	28.1	39.0
ChrX	513.5	429.1

If required, we can provide this information as a supplementary table. However, to try to further strengthen our analysis, we tried to perform OxoG staining in immunofluorescence with antibody Anti-8-Oxoguanine Antibody, clone 483.15 (MAB3560; Sigma-Aldrich) but unfortunately we failed to produce publication quality results.

4. Line 316-322. I see the data for the bulk based sequencing to assign the mutations with oncogenic potential in supplemental tables 12-16, but don't see a reference to where the data from the sequencing of individual clones is presented used to assign the hierarchies depicted in 6A-D. It seems like this data should be in the supplement, but I am unable to find it. How does the reconstruction by mutational analysis of individual colonies compare to the assignment that would have been made based on VAF?

We thank the Reviewer for highlighting this problem, however, following the specific requests of Reviewer #3, we completely removed this part of our study.

Minor-

5. Line 53- synthesize (spelling)

Corrected

6. Lines 71-75- run on sentence

Corrected

7. Figures 1C-E. While it is easy to appreciate that there are areas of low electron density in the mutants it is not easy to appreciate that the WT mitochondria have a rounder shape. Maybe just omit that statement, or alternatively present a measurement of circularity. I believe that the level of magnification is consistent between the images, but the scale bar is a bit challenging to see (especially in E), perhaps make the scale bar more prominent.

We thank the Reviewer for this suggestion. We added a measure of circularity, which confirms that WT mitochondria have a rounder shape. We also modified the scale bar of the figure in order to improve readability.

8. Line 124 and 138. I don't understand what is meant by vicariate in this particular context.

We rephrased to clarify our statement.

9. Figure S3. Can the font size be increased on the legends? What do the green bars represent?

Bars are now more visible and font size increased.

10. Figure S4. A brief description of the method used in the figure legend would be helpful.

A brief description was added to the previous legend.

11. Line 203. It would be more accurate to say that it restored to near normal levels

The text was modified as suggested.

12. Line 218. I would soften the statement that it “almost completely restored the normal ROS levels” perhaps to “reduced ROS levels to an intermediary level” or something similar. The levels of ROS are still fairly elevated relative to baseline.

We agree. Text was modified as suggested.

13. Line 243. A sentence or two explaining the 6-TG assay would be helpful here.

We thank the Reviewer for this suggestion. A description of the 6-TG assay was added.

14. Line 250. I may just be misreading this section, but if the 6-TG resistance assays were performed with P-ET and tigecycline as new assays from those described in line 245 they should be shown separately and not as a combined graph ie untreated controls should be from the same experiment. It looks like the control data is identical between supplemental figure 4 and 5...so I can't tell why they are presented as two different tables, unless the untreated is being shown for reference, but was not part of the second experiment.

This is correct: we performed a single assay with all the lines and treatments in a single experiment. To avoid any possible confusion we rephrased the entire paragraph.

15. Line 266. What is the evidence for aCML being characterized by chromosomal instability? A reference would be useful here.

A reference was added (Blood. 2014 Apr 24;123(17):2645-51).

16. 366-374 The discussion of the time involved to do functional studies on newly identified mutations does not seem necessary to me.

The paragraph focused on functional studies was deleted.

17. 396-397. I believe that it would be more correct to say “dominant” than “dominant negative”. Dominant negative implies that the mutant protein interferes with the activity of WT ETNK1, which I don't think has been demonstrated here.

In agreement with the Reviewer, “dominant negative” was changed to “negative”.

Answers to Reviewer #3

Summary of the work:

The authors have evaluated the effect of mutations in the kinase ETNK1 on mitochondrial activity in this paper. Using CRISPR KO and knock-in 293 HEK cell lines, they show that loss of ETNK1 has no significant effect on phospholipid levels in the mitochondria or in the

cell membrane, suggestive of mechanisms compensating for loss of ETNK1 enzymatic activity. However, they note that ETNK1 loss leads to increased mitochondrial mass, ROS production and Oxo-DG and γ H2AX, which is hypothesized to be due to a regulatory mechanism of mitochondrial complex II by phosphoethanolamine. Importantly, they demonstrate that the increased ROS and Oxo-DG in ETNK1 mutant cells is reversible by p-ET addition.

Overall I found the discoveries to be interesting, and relevant/ topical. However, the findings need to be defined further in a suitable cellular model, the link between ETNK1 loss and DNA damage needs to be clarified, and the presentation of data needs to be significantly improved in its rigour/ attention to methodological detail.

Major comments:

1. The majority of the experiments are based on ETNK1 knockout/knockin in 293 HEK cells. Given the significant differences in phenotype of mutations in distinct cellular contexts, I think it is required to carefully validate all critical findings in a leukemic or haematopoietic cell line, to justify the conclusion of ETNK1 regulating leukemogenesis through metabolism. The overexpression experiments in TF1 cells suggest that mutant ETNK1 can act as a dominant negative, but there is no data presented on the extent of overexpression that leads to this effect (the mutations in aCML are noted to be heterozygous).

We thank the Reviewer for this important observation, which was made also by the Reviewer #1. Indeed, the use of a single, non hematological cell line may lead to unreliable results. To overcome this limitation, in the revised version of our work we extended the use of the myeloid TF-1 cell line to all the experiments, confirming our previous findings (Fig 2, Suppl Fig S11, S12, S13, S19, S22 and corresponding text). Also, whenever possible taking into account the rarity of the primary samples, we extended the study to primary ETNK1+/- aCML samples (Fig 5, Suppl Fig S8, S20). Specifically, we replicated on the myeloid TF-1 line the following experiments:

- mitochondria activity with MitoTracker Red and Green in absence/presence of P-Et;
- mitochondria activity with JC1 plus/minus CCCP;
- ROS production in absence/presence of P-Et;
- γ H2AX Double-strand DNA damage in absence/presence of P-Et;
- mitochondria Complex I, II, III and IV activity in absence/presence of increasing concentration of P-Et;
- succinate competition assay for Mitochondria Complex II.

On primary samples we performed the following new analyses:

- membrane Lipid class composition analysis;
- ROS production in absence/presence of P-Et;
- γ H2AX Double-strand DNA damage in absence/presence of P-Et.

2. The cell line models are not adequately validated. Do the authors see a change in p-ET in their 293 KO/ knockin cells, similar to their previous observations? These data should be presented along with actual FISH analysis, sequencing, and western blots to show validation of the knockout and knockin in the 293 cell line model (along with validation of the overexpression system in the haematopoietic line).

This is an extremely important point. While in the past we showed that the presence of ETNK1 mutations in primary cells causes a decrease in the intracellular concentration of P-Et (Blood. 2015

Jan 15;125(3):499-503), here we didn't show the same finding on our CRISPR 293 model. The demonstration that in the CRISPR model cells show a similar behavior is of the utmost importance. To address this point, we generated new metabolomics assays, showing that our previous findings also apply to our 293-based CRISPR model and this is true both for the knock-in and the knock-out lines (Suppl. Fig S10).

To formally demonstrate the presence of the described variants, either knock-in or knock-out, we also generated new Sanger sequences, showing the presence of the expected genotypes. We also tried to generate publication quality Western blots of our CRISPR lines with several commercial anti-ETNK1 antibodies (Ethanolamine kinase (U-23): sc-130754, SANTA CRUZ BIOTECH, INC. and ETNK1 Polyclonal Antibody: A58978, Epigentek). Unfortunately, the quality of these antibodies is so poor that we are not able to provide the requested blot. To overcome, at least in part, this limitation, we performed Q-PCR experiments showing that ETNK1 is expressed at similar transcript levels in the WT and N244S lines (Suppl. Fig. S2; $p > 0.05$), while it is four-fold less expressed in the knock-out line compared to the WT one (Suppl. Fig. S2; $p = 0.0033$). This finding is compatible with an expected nonsense-mediated RNA decay occurring in the knock-out.

3. For all analysis represented in the paper, the authors are requested to specify the imaging/ quantification technique used, and the number of cells analyzed per experiment. Eg. Mitochondrial mass estimation has been performed with mitotracker green, but the methods and figure legend do not elaborate how the analysis was performed. The authors should also provide a table with all the antibodies used (clone/ product numbers), and concentration of primary antibody for each application.

We thank the Reviewer for this observation. We modified manuscript and figure legends accordingly. A table summarizing name, clone, brand, code, and dilution of used antibodies for each assay was added in the revised version (Suppl. Tab. S15).

4. ETNK1 mutations are hypothesized to reduce p-ET levels, leading to reduced “brakes” on SDH activity, and therefore increased ROS production from complex II, and subsequent DNA damage. Is there evidence of increased complex II activity in lysates from the ETNK1 KO/ knockin cells? Do these cells show evidence of altered respiration in a seahorse assay?

The altered respiration, as mitochondrial respiratory capacity and oxygen consumption (OCR), was measured in ETNK1-WT and ETNK1-KO intact cells using high-resolution respirometry Oroboros O2K (Figure 3A). While the argument is definitely complex and difficult to address, the complementary alteration of glucose uptake and lactate production (ECAR) was not measured with Seahorse XF since the results from metabolic profiling suggest this as the most likely explanation, at least in our opinion. The intracellular lactate levels are significantly lower in N244S cells, suggesting more carbon shunted to OXPHOS. In addition, lactate excretion is significantly lower in these cells, consequently ECAR (as one would measure in a seahorse assay) is lower. Unfortunately, during the procedure of mitochondria complex pull-down, cells are lysed and intracellular P-Et lost, thus levelling the differences across phenotypes and impairing our ability to perform the comparative assay of complex II activity.

5. If ROS generation is key to ETNK1 mutations in leading to genomic instability, does reduction of ROS reverse it? Would quenching of ROS (eg. by N-acetyl cysteine) reduce the amount of Oxo-DG and/or gH2AX? This would be pertinent given recently described links

between SDH mutations and impaired homologous recombination, as alternate means of leading to elevated γ H2AX/ genomic instability.

This is another extremely important point. To test the Reviewer's suggestion, we performed a quenching experiment with N-acetyl cysteine, using the measurement of the γ H2AX as a readout for the DNA damage. We found that, in the presence of N-acetyl cysteine, the number of γ H2AX foci in both mutated and knock-out lines decreases down to a level comparable to the WT, which suggests that indeed the mechanism responsible for the induction of DNA damage depends on the accumulation of ROS (Suppl. Fig. S17). On the contrary, we show that, as expected, exposure to UV light causes a significant increase in the γ H2AX foci in all cells, with WT reaching a level comparable to the mutated ones (Suppl. Fig. S18).

Globally, we think that this new data significantly strengthen the proposed model.

6. The section on ETNK1 being an early/ truncal event in leukemogenesis does not fit into the current flow of the paper. It could serve as an introductory section if required, or could be removed from this paper.

We thank the Reviewer for pinpointing this problem. We removed this part of the manuscript, as suggested.

Specific comments:

1. The Figure 1 title reads "Mitochondria morphology and activity". The authors use mitotracker green/red ratio for mitochondrial potential. As this is a key message of the paper, it would be good to cross-validate these findings with an orthogonal assay such as TMRE or JC1 fluorescence, in both the CRISPR KO and overexpression system. Positive and negative controls are important to include for these experiments, for example FCCP/CCCP treatment to demonstrate depolarization (Figure 1 and 2); ROS quenchers and inducers (Figure 3B).

This is an important point, as the alteration of mitochondrial potential in ETNK1-mutated samples is one of the key elements of this work. Following the Reviewer's suggestions, we cross-validated our findings by adding an orthogonal assay based on JC-1, which we tested in both 293 and TF-1 lines (Suppl. Fig. S3, S12).. Also, we added the missing positive and negative controls (Suppl. Fig. S14, S15), thanks for highlighting.

All the new experiments confirmed our previous findings, thereby strengthening our original hypothesis.

2. In Figure1 (panel C, D, E) authors show TEM images of mitochondria suggesting an abnormal mitochondrial morphology in ETNK1 mutant and KO cell lines. Are these accompanied by/ due to changes in cellular morphology?

We didn't notice any relevant change in cell morphology in ETNK1-WT, mutated, or KO lines. This likely suggests that the effect is restricted to mitochondria.

3. In figure 2 (and throughout the paper), please report actual p-values when t-tests are done. How many cells were analyzed for each experiment and using what method?

P-values and statistical tests are now explicitly reported in the manuscript, as well as the number of cells used for each experiment.

4. How does the in-vitro dose of Tigecycline of 2.5 microM compare to the pharmacokinetics of Tigecycline dosage in humans?

- Tigecycline Plasma Cmax = 3000ng/ml (Clin Pharmacokinet. 2009;48(9):575-84;.

- tigecycline MW = 585 g/mol;

*- therefore, Plasma Cmax = $3 * 10E-3g/l / 585g/mol = 5 * 10E-6 mol/l = 5 umol/l = 5uM$.*

The tigecycline plasma Cmax is 5uM, therefore the in-vitro dose of tigecycline (2.5uM) is likely to be compatible with drug pharmacokinetics in humans.

5. In figure 3, why were only WT and KO cells (and not the mutant) chosen for the O₂ consumption experiments? How many cells were analyzed per experiment for the CellRox experiments? How were the imaging data quantified?

For the O₂ consumption experiments we decided to choose only WT and KO cells to focus on the most extreme models, i.e. absence/presence of the ETNK1 protein.

At least 200 cells per experiment and condition were analyzed for the CellRox experiments, as well as for all the other confocal experiments. Confocal microscopy fields were analyzed using specific homemade-designed macro with ImageJ (<https://imagej.nih.gov/ij/>) software. Mitochondrial activity quantification was performed measuring the integrated density (ID) of red over green signal ratio, after proper manual thresholding. ROS production was analyzed measuring the ID in nuclear compartment. For the γ H2AX assay, the number of foci was detected after manual thresholding and with the precast "analyzed particle" plug-in of ImageJ. All the data obtained derived from at least ten fields per experimental condition.

6. Does ETNK1 loss/ mutation lead to increased overall OxoDG staining (as assessed by immunofluorescence)? For the 6TG assay (Figure 4D); please show representative images in the main or supplementary data

We tried to perform OxoDG staining in immunofluorescence with antibody Anti-8-Oxoguanine Antibody, clone 483.15 (MAB3560; Sigma-Aldrich) but unfortunately we failed to produce good quality results.

We now show the images of the 6TG experiments in the supplementary material (Suppl. Fig. S16) as suggested.

7. γ H2AX is smaller than Actin, and ideally should be presented below it in a western blot figure. The quality of the blots is not satisfactory (especially for in-vitro samples).

We shifted the position of γ H2AX/Actin in both panels of figure 5 (panels D and H) and we improved the quality of the image, as requested.

8. Figure 5B is not representative at present. Please highlight an area from figure 5A and enlarge for clearer view.

To improve the readability of figure 5B, we generated new, high resolution single cell images of both WT and KO lines.

9. How were three replicates performed for figure S10 and 5I (single patient sample)? Were they performed at three different occasions or processed simultaneously? How was the p-value calculated here?

They were three independent replicates processed at the same time point. The p-value was calculated using GraphPad Prism6 software, comparing primary bone marrow cells in the absence and presence of phosphoethanolamine 1mM.

10. “side pathways” could be phrased better

Changed to “alternative pathways”.

We sincerely hope we addressed all Reviewer' points and the revised manuscript is now suitable for publication in Nature Communications.

REVIEWER COMMENTS

Reviewer #1 (Remarks to the Author):

All my original concerns have been addressed

Reviewer #2 (Remarks to the Author):

The authors have thoughtfully responded to the reviewers and have satisfied all of my concerns. The addition of studies in TF1 cells enhances the relevancy of the work to myeloid malignancies. Overall this is an interesting and well-conducted study and I recommend publication.

Reviewer #3 (Remarks to the Author):

It is good to note that several of my previous points in relation to a suitable cellular model, assays used, ROS and DNA damage have been addressed in this revised version. The manuscript has improved significantly since the previous submission. I have a few comments that need to be addressed to the satisfaction of the handling editor.

1. Without western blots to confirm the knockout or degree of overexpression of ETNK1, it is difficult to comment on the validity of the experiments presented in the paper. I appreciate the difficulty posed by the current pandemic on acquiring reagents, but unfortunately this remains a key control experiment that should be performed.
2. The authors have improved the explanations of imaging/ quantitation technique used, but the data presentation and reporting still remains inadequate. Particularly in descriptions of the number of cells analyzed per experiment, or the number of replicates of each experiment (eg. For figure 1, how many cells were analyzed; how uniform is this phenotype?). The data representation checklist specifies that individual data points need to be represented, but this is not done through the manuscript- and it is unclear if the values presented in most figures come from technical replicates (cells) or biological replicates (experiments). Bar graphs do not convey the distribution of small datasets and should not be used. Dot plots, violin plots, or estimation plots are more appropriate.
3. Along the same lines, the statistical analyses used need to be appropriate. ANOVAs only state that there is a significant difference between multiple conditions; what was the post-correction used?

Please see for eg:

<https://www.nature.com/articles/s41592-019-0470-3>

<https://www.ncbi.nlm.nih.gov/pubmed/31929523>

<https://www.ncbi.nlm.nih.gov/pubmed/30917112>

<https://www.ncbi.nlm.nih.gov/pubmed/17420288>

Anand D Jeyasekharan
National University of Singapore

REVIEWER COMMENTS

Reviewer #1 (Remarks to the Author):

All my original concerns have been addressed

We thank Reviewer 1 for all his constructive and insightful comments.

Reviewer #2 (Remarks to the Author):

The authors have thoughtfully responded to the reviewers and have satisfied all of my concerns. The addition of studies in TF1 cells enhances the relevancy of the work to myeloid malignancies. Overall this is an interesting and well-conducted study and I recommend publication.

We thank Reviewer 2 for considering our work interesting and worth of publication in Nature Communications. We also wish to thank him for all his comments and constructive criticisms.

Reviewer #3 (Remarks to the Author):

It is good to note that several of my previous points in relation to a suitable cellular model, assays used, ROS and DNA damage have been addressed in this revised version. The manuscript has improved significantly since the previous submission. I have a few comments that need to be addressed to the satisfaction of the handling editor.

We are glad that Reviewer 3 considers the revised version of our manuscript significantly improved since the previous submission and that we were able to address several of the original criticisms.

1. Without western blots to confirm the knockout or degree of overexpression of ETNK1, it is difficult to comment on the validity of the experiments presented in the paper. I appreciate the difficulty posed by the current pandemic on acquiring reagents, but unfortunately this remains a key control experiment that should be performed.

1. We agree that the analysis of target protein expression is of great importance in order to test the validity of the subsequent experiments: as a matter of fact, during our previous revisions we made significant efforts to try to generate publication grade western blots. In the past these efforts proved to

be unsuccessful, owing to the absence of good or even decent quality anti-ETNK1 antibodies. To try to overcome this problem we tried here two different approaches:

A) We tried another round of western blot experiments by using a new anti-ETNK1 antibody recently proposed on the market (*GeneTex GTX105887*). Luckily, the quality of this third antibody proved to be better than all the previous ones: indeed, by loading a high amount of cell lysate, we managed to detect the expression of ETNK1 both in TF-1 cells overexpressing WT and mutated ETNK1 as well as in the empty TF-1 cells where, not unexpectedly, only a very weak band could be observed. This information is now part of the work and western blot data are now shown in supplementary figure 11. Unfortunately, by keeping the same blotting conditions, i.e. by loading a very high amount of cell lysate, we were not able to detect ETNK1 in 293 cells, which is likely explained by the very low expression level of ETNK1 in this line. It is also important to note that there is no evidence (e.g. presence of a degron region/PEST domain in the ETNK1 mutation hotspot) indicating that missense ETNK1 mutations should affect the stability of the protein. This point is also corroborated by the new blot (S11) where no significant difference in protein expression is detectable in WT vs mutated ETNK1 in the TF-1 line.

B) We performed mass spectrometry experiments in order to identify and quantify peptides unequivocally belonging to WT and mutated ETNK1. Unfortunately, although in principle highly specific peptides can be generated from full-length ETNK1 protein, the very weak constitutive expression of human ETNK1 in 293 cells prevented us from detecting them in our samples, while TF-1 overexpressing cells generated a single, specific peptide.

2. The authors have improved the explanations of imaging/ quantitation technique used, but the data presentation and reporting still remains inadequate. Particularly in descriptions of the number of cells analyzed per experiment, or the number of replicates of each experiment (eg. For figure 1, how many cells were analyzed; how uniform is this phenotype?). The data representation checklist specifies that individual data points need to be represented, but this is not done through the manuscript- and it is unclear if the values presented in most figures come from technical replicates (cells) or biological replicates (experiments). Bar graphs do not convey the distribution of small datasets and should not be used. Dot plots, violin plots, or estimation plots are more appropriate.

2. We thank Reviewer 3 for highlighting this important point. In this revised version of our manuscript we provided a more detailed description of our experiments, in particular in terms of number of cells analyzed in each experiment and number of replicates. We also changed all the histograms in main as well as supplementary figures, converting them to boxplots, as we agree that boxplots better convey the global distribution of our data. To also allow a direct investigation of all the experimental points, we reported individual values as dot-plots within each boxplot. We are confident that this approach will improve the overall readability of our experiments, also allowing to appreciate the detailed distribution of all data.

3. Along the same lines, the statistical analyses used need to be appropriate. ANOVAs only state that there is a significant difference between multiple conditions; what was the post-correction used?

In our tests we used Tukey's post-hoc tests, in order to better control for type I errors. This information is now reported in the manuscript and legends, as appropriate.

REVIEWERS' COMMENTS

Reviewer #3 (Remarks to the Author):

The authors have significantly improved the presentation of data in the paper. I have no further comments; pleased to recommend publication.

In the setting of open review (which I support to improve collegiality in science), it is not necessary to maintain the neutral term "the reviewer" when addressing comments.

Anand Jeyasekharan
National University of Singapore

REVIEWERS' COMMENTS

Reviewer #3 (Remarks to the Author):

The authors have significantly improved the presentation of data in the paper. I have no further comments; pleased to recommend publication.

In the setting of open review (which I support to improve collegiality in science), it is not necessary to maintain the neutral term "the reviewer" when addressing comments.

Anand Jeyasekharan
National University of Singapore

Response

We wish to thank Dr Jeyasekharan for his fruitful and constructive comments and for considering our work interesting and worth of publication in Nature Communications.